# Information Maximization Perspective of Orthogonal Matching Pursuit with Applications to Explainable AI

**Aditya Chattopadhyay**[†][*]  **Ryan Pilgrim**[†][*]  **René Vidal**[‡]
[†]Johns Hopkins University, USA, {achatto1, rpilgri1}@jhu.edu
[‡]University of Pennsylvania, USA, vidalr@upenn.edu

## Abstract

Information Pursuit (IP) is a classical active testing algorithm for predicting an output by sequentially and greedily querying the input in order of *information gain*. However, IP is computationally intensive since it involves estimating mutual information in high-dimensional spaces. This paper explores Orthogonal Matching Pursuit (OMP) as an alternative to IP for greedily selecting the queries. OMP is a classical signal processing algorithm for sequentially encoding a signal in terms of dictionary atoms chosen in order of *correlation gain*. In each iteration, OMP selects the atom that is most correlated with the signal residual (the signal minus its reconstruction thus far). Our first contribution is to establish a fundamental connection between IP and OMP, where we prove that IP with random projections of dictionary atoms as queries "almost" reduces to OMP, with the difference being that IP selects atoms in order of *normalized correlation gain*. We call this version IP-OMP and present simulations indicating that this difference does not have any appreciable effect on the sparse code recovery rate of IP-OMP compared to that of OMP for random Gaussian dictionaries. Inspired by this connection, our second contribution is to explore the utility of IP-OMP for generating explainable predictions, an area in which IP has recently gained traction. More specifically, we propose a simple explainable AI algorithm which encodes an image as a sparse combination of semantically meaningful dictionary atoms that are defined as text embeddings of interpretable concepts. The final prediction is made using the weights of this sparse combination, which serve as an explanation. Empirically, our proposed algorithm is not only competitive with existing explainability methods but also computationally less expensive.

## 1 Introduction

**Information Pursuit** (IP), first proposed by Geman and Jedynak [1], is a classical algorithm for active testing: Given a set of queries whose answers are informative about some target variable, IP proceeds by adaptively selecting queries in order of *information gain*. Specifically, in each iteration, IP selects the query whose answer has maximum mutual information about the target variable given the history of query-answers observed so far. IP has found numerous applications in machine learning such as tracking roads in satellite images [1], face detection & localization [2], detecting & tracking surgical instruments [3], and scene interpretation [4]. More recently, IP has been touted as a framework for explainable AI [5]. Despite its utility, a major limitation for IP is its reliance on mutual information, which is challenging to compute in high dimensions [6]. In light of this, we ask the question, are there alternative sequential algorithms that could replace IP by employing a simpler objective in their selection step? One such algorithm is Orthogonal Matching Pursuit (OMP), which we describe next.

---

[*]Equal contribution

**OMP** [7, 8] is a classical algorithm for sparse coding: given a dictionary of atoms and an observed signal, OMP proceeds by adaptively selecting atoms in order of *correlation gain*. Specifically, in each iteration, OMP selects the atom that has maximum correlation with the current residual (the unexplained part of the observed signal given the previously selected atoms). OMP has been successfully applied to many signal processing applications such as image denoising using over-complete dictionaries [9], image super-resolution [10, 11], and matrix completion [12].

**Similarities between IP and OMP.** Although the two algorithms were discovered independently in two different communities, IP in the active testing and OMP in the signal processing community, they share many similarities. Both algorithms are illustrated in Figure 1.

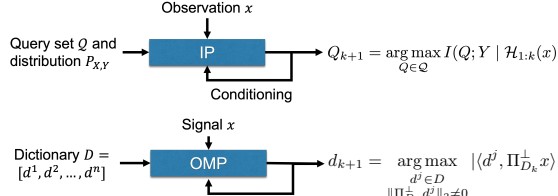

- Both seek a parsimonious representation. IP tries to make a prediction by selecting the minumum number of queries on average. OMP tries to represent the observed signal by selecting the minimum number of atoms.

Figure 1: **IP and OMP.** Both are greedy sequential algorithms. IP selects the next query using conditional mutual information $I(\cdot\,;\cdot)$ while OMP selects the next dictionary atom based on dot products $\langle\cdot,\cdot\rangle$. Previous choices are incorporated via the history $\mathcal{H}_{1:k}(x)$ of previous query answers in IP, and the projection matrix $\Pi_{D_k}^{\perp}$, which we define in § 2, in OMP.

- Both are greedy algorithms used as efficient approximations to the true optimal solution.[2] In each iteration, IP uses conditional mutual information to decide the next most *informative* query about the target variable $Y$. Similarly, in each iteration, OMP uses the dot product between a dictionary atom and the residual (the observed signal's projection onto the orthogonal complement of the span of all the atoms selected so far) to select the next most *representative* atom to reconstruct the observed signal. The conditioning (in IP) and residual (in OMP) are mechanisms to account for previous decisions made by each algorithm.

- IP terminates when the posterior $P(Y \mid \text{History})$ is sufficiently peaked, indicating $Y$ can be predicted with low error. Similarly, OMP terminates once the mean squared error between the observed signal and its reconstruction is sufficiently low, indicating that the observed signal can be reconstructed from the selected atoms with high fidelity.

**Differences between IP and OMP.** Given these similarities, one may be tempted to conjecture that one of the algorithms is a particular case of the other. However, establishing such a connection is not trivial at all. The reason is that there are fundamental differences in the manner in which these two algorithms operate, despite their apparent similarities. In IP, the target variable and query answers are all random variables. An illustrative example of this is given by the popular parlor game "twenty questions", where a player asks queries about attributes of some entity $Y$ (that another player has thought of) and would like to identify $Y$ by asking the minimum number of questions. In this example, $Y$ is a random variable, and as a result, the corresponding query answer which depends on $Y$ is also a random variable. In OMP, on the other hand, one typically observes some fixed signal and then proceeds to encode it using the dictionary atoms. Consequently, IP's objective to select the next query requires estimating mutual information, which, as already stated, is difficult in high dimensions. OMP, on the other hand, only relies on dot products and least-squares involving the fixed signal and small numbers of dictionary atoms, which is computationally a far more tractable objective. Therefore, it is indeed surprising that one can derive a connection between the two algorithms.

**Is OMP a particular case of IP?** Formally establishing a connection between IP and OMP requires answering two key questions: (i) "what would the queries and their corresponding answers be?" and (ii) "what would the target variable be?" As IP operates on random variables and OMP on fixed observed signals, we need to inject the observed signal into the definition of the queries and/or target variable. We achieve this by taking queries as dot products of the dictionary atoms with a standard normal variable $Z$, and the target variable as the dot product between between the observed signal $x$ and $Z$. The intuition is that since $Z$ has a radially symmetric distribution, in order to accurately

---

[2]In the case of IP, this is the solution that needs the minimal number of queries on average to predict the target variable with a desired level of confidence; in the case of OMP, this is the sparsest code that reconstructs the observed signal up to a desired level of accuracy.

reconstruct the target dot product, IP would need to encode the observed signal into the selected queries (scalar projections of the dictionary atoms). We formally show this in §3. In particular, we show that IP selects the atom in each iteration whose projection (onto the orthogonal complement of the span of the atoms selected so far) has the maximum normalized dot product with the residual. This contrasts with OMP which uses the unnormalized dot product as its selection criterion. We call this IP version of sparse coding *IP-OMP*. We then show via simulations over random Gaussian dictionaries and different sparsity levels that despite this algorithmic difference, IP-OMP and OMP have almost identical sparse recovery success rates. This empirical observation is further complemented by theoretical work by Soussen et al. [13] who prove that if for a given dictionary, exact recovery for all $s$-sparse signals is possible after $s$ iterations of OMP, then the set of atoms selected after termination ($s$ iterations) would be the same for both IP-OMP and OMP.[3]

**Implications of this connection.** As alluded to before, computing mutual information is often intractable in high dimensions. Previous approaches tackle this issue by either explicitly learning a generative model for data [5, 14] or by learning the most informative next query from data using deep networks and stochastic objectives [15, 16]. However, IP-OMP presents a much simpler alternative which does not involve training any deep networks or learning data distributions using generative models. This begs the question of whether IP-OMP can be used as a cheap surrogate to IP in certain applications. In this paper, we show one such application to explainable AI, where IP has been recently proposed as a method for making explainable predictions by design [15]. More specifically, [15] proposes a framework called Variational Information Pursuit (V-IP) where the query set $\mathcal{Q}$ consists of interpretable queries about the data. Using $\mathcal{Q}$, V-IP explains its prediction in terms of the selected interpretable queries. Similarly, we construct a dictionary comprised of interpretable atoms from text embeddings. We then propose to use IP-OMP to represent the image in terms of a sparse combination of dictionary atoms (the sparse code). The final prediction is made by training a linear classifier on top of the sparse code. In this way, we can make predictions explainable—in the sense that humans can understand predictions in terms of the sparse interpretable inputs to the classifier and their corresponding weights—without computing mutual information. This, however, presents a challenge: images will typically not be represented well as linear combination of text embeddings since they are very different modalities. To remedy this, we propose to use CLIP [17], a recently introduced large Vision-Language Model, to encode both images and text into a shared latent space.

**Paper contributions.** To summarize, our main contributions are:

- We formally establish a connection between IP and OMP. By choosing the parameters of IP appropriately, we obtain an algorithm, IP-OMP, which is equivalent to OMP up to a normalization in the objective.

- Inspired by this connection, we propose a simple algorithm using IP-OMP and CLIP for making explainable predictions in visual classification tasks. We show empirically on multiple image classification datasets that the performance of our algorithm is competitive with, if not better than, state-of-the-art methods for explainable AI.

## 2 Preliminaries

### 2.1 Information Pursuit

In this paper, random variables are defined over a common sample space $\Omega$ and are denoted by capital letters. Their realizations are denoted by the corresponding lowercase letter. Let $Y$ be a target variable of interest. Let $\mathcal{Q}$ be a set of queries, where every query $Q \in \mathcal{Q}$ is a random variable. We call the realization of $Q$, written $q$, its answer. Given $\mathcal{Q}$, one is often interested in predicting $Y$ by sequentially asking queries from $\mathcal{Q}$ such that the average number of queries needed is minimized [18, 19, 20, 14, 5, 16]. Information pursuit (IP) [1] greedily approximates the solution to this problem. In particular, a single IP run proceeds as follows,

$$Q_1 = \arg\max_{Q \in \mathcal{Q}} I(Q; Y); \quad Q_{k+1} = \arg\max_{Q \in \mathcal{Q}} I(Q; Y \mid \mathcal{H}_{1:k}), \tag{1}$$

where $Q_{k+1}$ is the query selected in iteration $k + 1$, $I(\cdot\,;\cdot)$ denotes mutual information and $\mathcal{H}_{1:k}$ is the history of query-answer pairs observed after the first $k$ iterations. More precisely, $\mathcal{H}_{1:k}$ is defined

---

[3]The authors of [13] refer to IP-OMP as the Orthogonal Least Squares algorithm since it can also be derived from a least squares optimization perspective. We expand upon this point in §3.

as the event $\{\omega \in \Omega : Q_1(\omega) = q_1, \ldots, Q_k(\omega) = q_k\}$. The algorithm terminates after $\tau$ iterations if the entropy (or differential entropy if $Y$ is a continuous random variable) of the posterior $P(Y \mid \mathcal{H}_{1:\tau})$ is below a user-defined threshold [1]. Alternatively, one could employ a fixed budget of $\tau$ iterations and terminate [21]. After termination, IP's prediction for $Y$ is given by $\arg\max_Y P(Y \mid H_{1:\tau})$ [5].

## 2.2 Orthogonal Matching Pursuit

Given a matrix $D \in \mathbb{R}^{m \times n}$ with $m \ll n$ and an observed signal $x \in \mathbb{R}^m$, the goal of sparse coding is to find a sparse vector $\beta \in \mathbb{R}^n$ that reconstructs the signal $x$. More precisely, the goal is to solve

$$\min_{\beta \in \mathbb{R}^n} \|\beta\|_0 \quad \text{s.t.} \quad x = D\beta, \tag{2}$$

where $\|\cdot\|_0$ refers to the $\ell_0$ pseudo-norm. Equation 2 has been of significant interest over the past few decades due to its applications to compressed sensing [22, Chapter 1] and sparse representation theory [23]. We will call $D$ the dictionary, $x$ the observed signal and $\beta$ the sparse code. We will refer to the columns of $D$ as atoms, where $d^j$ refers to the $j^{\text{th}}$ atom.

Since the problem in equation 2 is NP-Hard [22, §2.3], approximations are employed. Orthogonal Matching Pursuit (OMP) [7, 8] is a popular greedy algorithm known for its computational efficiency [24] compared to alternatives like Basis Pursuit [25]. Given $x$ and $D$, the OMP algorithm first $\ell_2$-normalizes all the atoms in $D$. Subsequently, it initializes its estimate of the sparse code as $\hat{\beta}_0 = 0$ and its estimate of the support of $\beta$ (set of non-zero indices in $\beta$) as the empty set $S_0 = \emptyset$. OMP then proceeds iteratively. Its $(k+1)^{\text{th}}$ iteration ($k \geq 0$) is defined as,

$$\begin{aligned}
j_{k+1} &= \arg\max_{j \in \{1,\ldots,n\}} |\langle d^j, x - D\hat{\beta}_k\rangle|; \quad S_{k+1} = S_k \cup \{j_{k+1}\}; \\
\hat{\beta}_{k+1} &= \arg\min_{u \in \mathbb{R}^n} \{\|x - Du\|_2 \ : \ \text{support}(u) \subseteq S_{k+1}\},
\end{aligned} \tag{3}$$

where $x - D\hat{\beta}_k$ is called the residual at iteration $k$ [22, §3.2]. There are multiple choices for the termination criteria, including (i) to run some fixed number of iterations $\tau$, or (ii) to terminate once the norm of the residual is below some user-defined threshold $\epsilon \geq 0$, that is, $\|x - D\hat{\beta}_\tau\|_2 \leq \epsilon$ [26].

We now rewrite the atom selection step in equation 3 into a form that is more amenable to comparison with the IP version of OMP called IP-OMP that we will derive next. For a particular run of the OMP algorithm, let $D_k$ be a sub-matrix of $D$ composed of the first $k$ atoms selected by the OMP algorithm and let $\Pi_{D_k}$ be the projection to the range of $D_k$. From the update of $\hat{\beta}_k$, it is clear that $D\hat{\beta}_k$ is a projection of $x$ onto the range of $D_k$. Similarly, the residual $x - D\hat{\beta}_k$ is the projection of $x$ onto the orthogonal complement [27]. Using this observation, we rephrase the atom selection in equation 3 as

$$j_{k+1} = \arg\max_{j \in \{1,\ldots,n\}} |\langle d^j, \Pi_{D_k}^\perp x\rangle| = \arg\max_{j \in \{1,\ldots,n\}} |\langle \Pi_{D_k}^\perp d^j, \Pi_{D_k}^\perp x\rangle|. \tag{4}$$

# 3 IP-OMP: Information Pursuit for Sparse Coding

Recall from §1 that to establish a connection between IP and OMP, we need to specify a set of queries and a target variable, all of which are random variables. Moreover, since OMP operates on deterministic inputs (the observed signal), we have to inject the observed signal into the definition of the queries and/or the target variable. Inspired from the Bayesian perspective of compressed sensing [28, 29] one natural attempt could be to consider a sparsity-inducing prior over the sparse code, like the Laplace distribution [30]. This induces a distribution over the observed signal $X = D\beta +$ measurement noise (capitalized to emphasize that $X$ is now a random variable). Since the goal is to sparse-code the signal, a natural choice for the target variable could be $X$ itself. Finally, since we want to recover the atom selection step in OMP (equation 4) using IP, which is obtained by maximizing the absolute dot product between the atom and the residual, a reasonable choice for the query set $Q$ could be atoms of the dictionary where the answer to a query is the corresponding atom's dot product with $X$. Given this setup, the first query selected by IP is[4]

$$Q_1 = \arg\max_{Q^j \in \mathcal{Q}} I(Q^j; X) = \arg\max_{d^j \in D} I(\langle d^j, X\rangle; X). \tag{5}$$

---

[4]Note that we use subscript $Q_i$ to denote the query selected by IP at iteration $i$, and superscript $Q^i$ to denote the query that corresponds to the $i^{\text{th}}$ atom, $d^i$, of $D$.

It is immediately clear from equation 5 that the first query would be independent of the observed signal $X = x$. This is because IP decides what is the most informative query solely based on mutual information, which is a property of the distribution, and does not involve observing the realization $x$. This is in sharp contrast to OMP, where the selection of the first atom is driven by the observed signal (see equation 3). Thus, assuming a generative model for the signal $X$ would not result in the OMP algorithm and calls for a more "non-standard" way of defining the queries and/or the target variable.

**What is the right query set and target variable?** How can we make the queries and/or the target variable depend on a realization of $X = x$? Inspired by a measure called sliced mutual information, which defines the dependence between two random variables as the average mutual information between their one-dimensional projections onto random directions [31], we propose the following query set and target variable.

- Define $Z$ to be a standard normal variable in $\mathbb{R}^m$. Let $z^0$ be a sample from $P_Z = \mathcal{N}(0, I_m)$.
- Take $\mathcal{Q}$ to contain all the atoms in $D$ as queries, such that query $Q^j \in \mathcal{Q}$ corresponds to the random projection $\langle d^j, Z \rangle$ of atom $d^j \in D$, and the corresponding answer is $q^j := \langle d^j, z^0 \rangle$.
- Take the target random variable of interest as $\langle x, Z \rangle$. Thus, given $z^0$, the value of the target variable that needs to be predicted is $\langle x, Z \rangle = \langle x, z^0 \rangle$.

The rationale behind these choices is as follows. Since $Z$ is a random vector with a radially symmetric density, to accurately predict the target variable $\langle x, Z \rangle$ for all realizations of $Z$, IP needs to query atoms that would in a way *code* for $x$ itself. This intuition is formalized by the following lemma.

**Lemma 1.** *Let $z^0$ be a realization of $Z$. Given a subset $\mathcal{L}$ of queries from $\mathcal{Q}$ (or equivalently, atoms selected from $D$), the conditional entropy of the target $\langle x, Z \rangle$ given the query answers observed is*

$$h\Big( \langle x, Z \rangle \mid \{Q = \langle d_Q, z^0 \rangle : Q \in \mathcal{L}\} \Big) = \frac{1}{2} \ln(2\pi e \|\Pi_{D_\mathcal{L}}^\perp x\|_2^2), \tag{6}$$

*where $h$ is the conditional differential entropy, $d_Q$ denotes the atom used to construct the random variable $Q$, $D_\mathcal{L}$ is the sub-matrix of $D$ consisting of atoms corresponding to queries in $\mathcal{L}$ and $\Pi_{D_\mathcal{L}}^\perp$ is the projection matrix onto the orthogonal complement of the range of $D_\mathcal{L}$.*

The proof of this lemma relies on the joint Gaussianity of all the query answers and the target variable since they are all dot products with the same normal vector $Z$. Proof in Appendix §A.1.

Notice that the history of query answers observed, after a finite number of iterations of IP, is just some subset of queries selected from $\mathcal{Q}$ along with their corresponding answers. Since IP greedily selects queries until the conditional entropy of the target (given history) is minimized (§2.1), Lemma 1 indicates that IP would terminate when $\|\Pi_{D_\mathcal{L}}^\perp x\|_2^2 \to 0$, that is, when $x$ can be "almost exactly" reconstructed from the atoms selected by IP.

**The IP-OMP algorithm.** Given this choice of query set and target variable, we now state our main result (Theorem 1) proving that IP not only chooses atoms that *code* for $x$, but also uses exactly the same query selection strategy as the atom selection strategy of OMP up to a normalization factor.

**Theorem 1.** *Given an observed signal $x$, a realization $z^0$ of $Z$, and the choice of $\mathcal{Q}$ and the target variable as described in this section, IP proceeds as follows for $k \geq 1$,*

$$Q_1 = \underset{Q^j \in \mathcal{Q}}{\arg\max}\, I(Q^j; \langle x, Z \rangle) = \underset{d^j \in D}{\arg\max}\, \frac{|\langle d^j, x \rangle|}{\|d^j\|_2 \|x\|_2}$$

$$Q_{k+1} = \underset{Q^j \in \mathcal{Q}}{\arg\max}\, I(Q^j; \langle x, Z \rangle \mid \mathcal{H}_{1:k}) = \underset{d^j \in D, \|\Pi_{D_k}^\perp d^j\|_2 \neq 0}{\arg\max}\, \frac{|\langle \Pi_{D_k}^\perp d^j, \Pi_{D_k}^\perp x \rangle|}{\|\Pi_{D_k}^\perp d^j\|_2 \|\Pi_{D_k}^\perp x\|_2}, \tag{IP-OMP}$$

*where $\mathcal{H}_{1:k} := \{Q_1 = \langle d_1, z^0 \rangle, Q_2 = \langle d_2, z^0 \rangle, \ldots, Q_k = \langle d_k, z^0 \rangle\}$, $d_i$ is the atom (query) selected by IP in the $i^{th}$ iteration, $D_k$ is the sub-matrix of $D$ consisting of atoms corresponding to queries selected by IP in the first $k$ iterations and $\Pi_{D_k}^\perp$ is the projection matrix onto the orthogonal complement of the range of $D_k$.*

The above result uses the joint Gaussianity of all the query answers and the target variable. A full derivation can be found in Appendix §A.2. Comparing equation 4 and IP-OMP, it is clear that the two algorithms differ by a normalization. To distinguish from the OMP algorithm, we call the

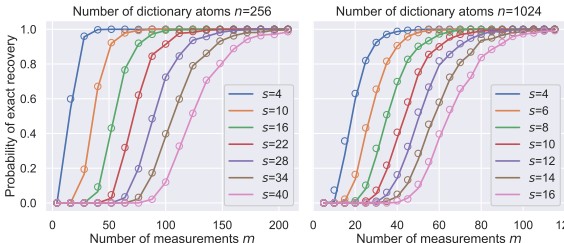

Figure 2: Performance of OMP (solid curves) and IP-OMP (overlaid circles) in the noiseless case as a function of the number of measurements $m$ for a fixed sparsity level $s$ and a number of dictionary atoms $n$. "Probability of exact recovery" reports the fraction of times (over 1000 simulations) the algorithm exactly recovered the sparse code for a particular setting of $(m, n, s)$.

iterative algorithm described above as the IP-OMP algorithm. If one uses the entropy of the posterior $P(\langle x, Z \rangle \mid \mathcal{H}_{1:\tau})$ criterion for termination after $\tau$ iterations, then according to Lemma 1 the algorithm would terminate once $\|\Pi_{D_\tau}^\perp x\|_2^2 \leq \epsilon$. This is reminiscent of the norm of the residual termination criterion commonly employed by OMP.

**Sparse coding using the IP-OMP algorithm.** In the discussion so far between the equivalence of OMP and IP-OMP, we swept the issue of recovering the sparse code under the rug. To remind the reader, the goal of sparse coding is to find a code $\hat{\beta}_{\text{IP-OMP}}$ such that $x \approx D\hat{\beta}_{\text{IP-OMP}}$. However, the IP-OMP updates do not explicitly provide a way of doing so. However, by analyzing IP-OMP's prediction of the target $\langle x, Z \rangle$ we can derive $\hat{\beta}_{\text{IP-OMP}}$.

Recall from §2.1 that after termination, IP's prediction is the mode of the posterior distribution. Thus after termination IP-OMP's prediction would be given by

$$\max_Z P(\langle x, Z \rangle \mid \mathcal{H}_{1:\tau}) = \langle D_\tau D_\tau^\dagger x, z^0 \rangle, \tag{7}$$

where $\tau$ is the number of iterations before IP-OMP terminated, $D_\tau$ is the sub-matrix of $D$ consisting of the atoms selected in the first $\tau$ iterations, $\dagger$ represents the pseudo-inverse, and $z^0$ is the realization of $Z$ that determines the outcomes of all the query answers and the target (which is $\langle x, Z \rangle = \langle x, z^0 \rangle$). Refer to Appendix §A.3 for a proof of equation 7. Next, notice that the query selection in IP-OMP is independent of the realization of $Z$, which implies that the sequence of queries selected in the first $\tau$ iterations would depend solely on $x$. This implies the following lemma,

**Lemma 2.** *For all $z \in \mathbb{R}^m$, IP-OMP's prediction of the target is given by $\langle D_\tau D_\tau^\dagger x, z \rangle$.*

Refer to Appendix §A.4 for a proof. Taking $z = e_i$, the $i^{\text{th}}$ canonical basis vector in $\mathbb{R}^m$, we observe that IP-OMP's prediction of the $i^{\text{th}}$ component of $x$, denoted as $x_i$, is $(D_\tau D_\tau^\dagger x)_i$. Hence, we conclude

$$x \approx D_\tau D_\tau^\dagger x = \overbrace{[D_\tau \; D_{-\tau}]}^{D} \overbrace{\begin{pmatrix} D_\tau^\dagger x \\ 0 \end{pmatrix}}^{\hat{\beta}_{\text{IP-OMP}}} \implies \hat{\beta}_{\text{IP-OMP}} = \begin{pmatrix} D_\tau^\dagger x \\ 0 \end{pmatrix}, \tag{8}$$

where $D_{-\tau}$ is the matrix comprised of atoms of $D$ not selected by IP-OMP in the first $\tau$ iterations. Comparing with equation 3, we see that this is analogous to the sparse code estimate OMP obtains after $\tau$ iterations, with the $D_\tau$ matrix now replaced with the atoms selected by OMP.

**IP-OMP $\equiv$ OMP.** Recall the only difference between IP-OMP and OMP is the normalization step after the dot product. We carry out synthetic experiments to investigate the effect of this normalization factor on the practical performance of IP-OMP vs. OMP in sparse code recovery. We first consider the noiseless case (equation 2) and later investigate the scenario where signals are corrupted by noise.

For the noiseless case, we conduct the following experiment. Fix $n$, the dimension of the sparse code, $m$ the dimension of the observed signal, and $s$ the sparsity level. Generate a large number of random dictionaries, where each atom (column) is sampled from the uniform distribution on the unit sphere in $\mathbb{R}^m$. For every dictionary, pick a subset of $s$ (out of $n$) atoms uniformly at random and simulate the coefficients for each of the selected atoms from the standard normal distribution. Set all the remaining coefficients to 0. These coefficients constitute the sampled sparse code. For each simulation, construct the observed signal $x_o = D_o \beta_o$, where $D_o$ is the sampled random dictionary and $\beta_o$ is the sampled sparse code. For every $x_o$, we carry out both OMP and IP-OMP and measure their performance in terms of the fraction of simulations in which the algorithm exactly recovered the sparse code $\beta_o$. We define exact recovery when the normalized mean-squared error (NMSE)

$$\text{NMSE}(\hat{\beta}_{\text{Alg.}}, \beta_o) = \|\hat{\beta}_{\text{Alg.}} - \beta_o\|_2^2 / \|\beta_o\|_2^2 \tag{9}$$

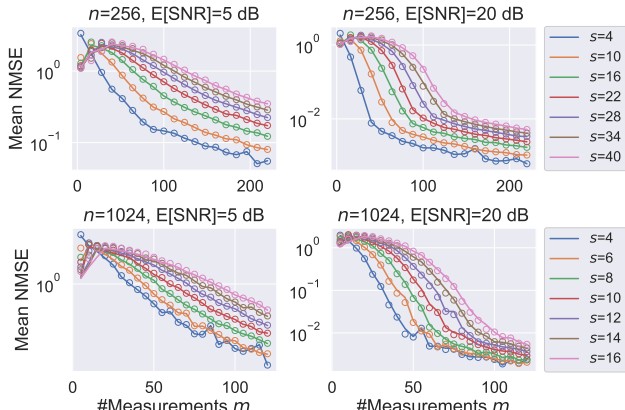

Figure 3: Performance of OMP (solid curves) and IP-OMP (overlaid circles) in the case where signals are corrupted by noise. On the $y$-axis we report the mean NMSE, as defined in equation 9, averaged over 1000 trials. On the $x$-axis, we report the number of measurements (signal dimension) $m$. Each sub-figure corresponds to a fixed number of dictionary atoms $n$ and a fixed expected signal-to-noise ratio $\mathbb{E}[\text{SNR}]$. The top row shows results for small $n$ (256), while the bottom shows results for large $n$ (1024).

is approximately zero ($< 10^{-14}$), where the subscript $_{\text{Alg.}}$ indicates the algorithm (OMP or IP-OMP). This experimental setup was inspired by [24]. Further details of the experiment can be found in Appendix §B. Figure 2 reports these results which indicate that OMP and IP-OMP achieve *almost* the same exact recovery rate over a variety of different values of $(m, n, s)$. In particular, both IP-OMP and OMP degrade similarly when the sparsity rate $\frac{s}{m} \to 1$ or the measurement rate $\frac{m}{n} \to 0$.

For the noisy case, we extend equation 2 with additive Gaussian noise. Specifically, we consider the measurement model $\widetilde{x} = D_o \beta_o + e$, where $D_o$ and $\beta_o$ are sampled as described in the previous paragraph, $\widetilde{x}$ is the noisy observed signal and $e \sim \mathcal{N}(0, \sigma^2 I)$. The noise variance $\sigma^2$ was set to achieve particular values of the expected signal-to-noise ratio (SNR)

$$\mathbb{E}[\text{SNR}] = \mathbb{E}_{D_o, \beta_o, e}\left[\frac{\|D_o \beta_o\|_2^2}{\|e\|_2^2}\right]. \tag{10}$$

Details on this calculation are provided in Appendix §B.3. We measure the quality of the estimates produced by OMP and IP-OMP by reporting the mean NMSE of their estimates, $\hat{\beta}_{\text{Alg.}}$ (taken over all the trials). Further details can be found in Appendix §B. Figure 3 shows that the the empirical equivalence between OMP and IP-OMP is maintained over different values of $(m, n, s, \mathbb{E}[\text{SNR}])$.

More results can be found in Appendix §B. Figure 2 and Figure 3 show that despite the extra normalization, IP-OMP and OMP have practically the same sparse code recovery rates for random Gaussian dictionaries, which are commonly studied in compressed sensing (see, e.g., [22, Ch. 9]).

**Connection to Orthogonal Least Squares (OLS).** Interestingly, the IP-OMP selection criterion (equation IP-OMP) results in greedily selecting the atom that maximally reduces the residual error [32]. Specifically, let $D_k$ indicate the set of atoms from $D$ obtained after $k$ iterations of IP-OMP. Then, the atom $d^*$ solving the following optimization problem

$$(d^*, \hat{\beta}^*) = \underset{d \in D, \hat{\beta} \in \mathbb{R}^{k+1}}{\arg\min} \left\| x - [D_k \ d]\hat{\beta} \right\|_2^2, \tag{11}$$

is equal to the $\arg\max$ of the IP-OMP objective at iteration $k + 1$. Using this insight, multiple authors have proposed to modify OMP by replacing its atom selection criterion with equation 11 and have re-discovered this algorithm under different names [33] with OLS being one of them. In this paper, we coin the name IP-OMP to emphasize its derivation from information-theoretic principles.

Our results on the empirical equivalence between IP-OMP and OMP are complemented by prior investigations into the OLS algorithm which report similar conclusions using random Gaussian dictionaries [34, 35]. Theoretically, Soussen et al. [13] showed that both the algorithms share the same exact recovery conditions. This is made precise in Theorem 2 (a restatement of results in [13]).

**Theorem 2.** *Given a dictionary $D$ and support $\mathcal{S}$, define $\mathcal{B} = \{\beta \in \mathbb{R}^n : \text{supp}(\beta) \subseteq \mathcal{S}\}$. Then IP-OMP recovers all $\beta \in \mathcal{B}$ in at most $s = |\mathcal{S}|$ iterations if and only if the same holds for OMP.*

These results further strengthen our claim that for practical scenarios, the normalization factor in IP-OMP does not have a significant impact. We note in passing that, for certain "special" dictionaries in which the dictionary atoms are strongly correlated with each other, IP-OMP has been reported to outperform OMP in its ability to recover the sparse code [13, 34].

# 4 IP-OMP for Explainable Predictions for Visual Classification Tasks

Inspired by the recent application of IP [5] and its variant V-IP [15] to explainable AI, here we propose a simple explainable AI algorithm for visual classification tasks which uses IP-OMP to ask queries about the image. In §4.1 we briefly describe the V-IP framework, propose our CLIP-IP-OMP algorithm, and discuss the differences with V-IP. Then, in §4.2 we present experiments showing that CLIP-IP-OMP is computationally much cheaper than V-IP, while achieving a similar classification accuracy. Refer to Appendix §C for a review of prior work on explainable AI.

## 4.1 The CLIP-IP-OMP Algorithm

**The V-IP algorithm.** In V-IP, a user first specifies a query set, which is comprised of interpretable functions of the input. For example, if the task is animal classification, the questions could be about various animal attributes. V-IP then adaptively generates short query-answer chains that are sufficient for prediction. This is done by learning a querier and a classifier network simultaneously from data. The querier learns to select the most informative query given the history of query-answer pairs obtained so far, while the classifier learns to make predictions based on the same history. However, the number of possible histories (subsets of query-answer pairs) the querier and classifier have to learn is exponential in the size of the query set, making the training process very slow. In this paper, we propose an alternative algorithm which is computationally much cheaper.

**The CLIP-IP-OMP algorithm.** Analogously to V-IP's interpretable query sets, we propose to use dictionaries of interpretable atoms. For example, every atom of the dictionary could correspond to a text embedding [36] of a semantic concept relevant for the classification task. This poses a challenge, since images and text embeddings live in very different spaces (pixel space vs. the latent space of some deep network). To address this mismatch, we leverage CLIP [17], which learns to encode images and text into a shared latent space such that the dot products of their respective embeddings are reflective of how well the text describes the image contents. We propose the following algorithm for making explainable predictions for visual classification tasks:

1. Given a set of semantic concepts relevant to the classification task, define each atom of the dictionary $D$ as the CLIP embedding of one of the concepts obtained using CLIP's text encoder.

2. For a given input image, define the observed signal $x$ as the CLIP embedding of the image.

3. Using IP-OMP as the querier, encode the image's CLIP embedding as a sparse combination of dictionary atoms with weights $\hat{\beta}_{\text{IP-OMP}}$. While both OMP and IP-OMP recover the sparse code similarly well, we choose to use IP-OMP in our experiments due to its nice information-theoretic interpretation (§3).

4. Train a linear classifier to predict the class label from the sparse codes $\hat{\beta}_{\text{IP-OMP}}$. This choice is made to enhance interpretability. By inspecting the weights of the linear layer, one can identify which concepts (atoms) are most important for the prediction.

For a given image, the linear network predicts the class label based on the sparse code. The explanation of the prediction, then, is given in terms of the non-zero coefficients of $\hat{\beta}_{\text{IP-OMP}}$ and their corresponding atoms. Because these atoms are text embeddings for concepts, they are semantically meaningful. This is in stark contrast to predictions made by deep networks trained on raw pixel data, where the reasoning behind network predictions is opaque.

**Differences between CLIP-IP-OMP and V-IP.** A significant difference between CLIP-IP-OMP and V-IP is the target variable in question. In V-IP, this is the class label $Y$, whereas in IP-OMP this is a projection of $x$ (the observed image's CLIP embedding) onto a random direction $Z$. Moreover, the query answers in both algorithms are different. In V-IP, the image is considered a random variable and answers are functions of the image. In IP-OMP, the answers are simply dot products between a query (dictionary atom) and the random direction $Z$. Despite these differences, the label prediction in both V-IP and CLIP-IP-OMP is explained using a small set of concepts selected before termination. In V-IP, this is the set of query-answer pairs selected. In CLIP-IP-OMP, this is the sparse code $\hat{\beta}_{\text{IP-OMP}}$, which encodes the set of selected concepts and their weights.

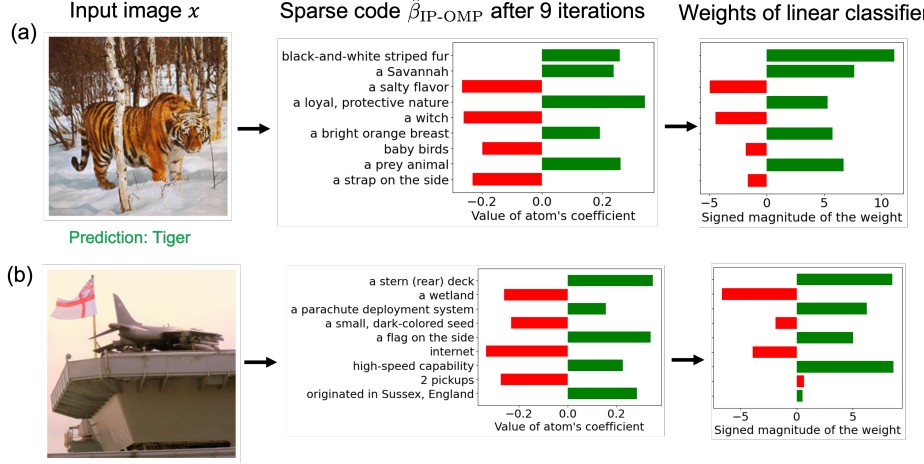

Figure 4: **Explainable predictions using CLIP-IP-OMP.** Two example runs of our algorithm on two images from the ImageNet test set, with number of iterations $\tau = 9$. For both images, our model predicted the correct class with $68.73\%$ & $72.21\%$ confidence resp. The weights of the linear classifier correspond to the weights for the predicted class. More examples in Appendix §E.6.

## 4.2 Experiments

Having described our algorithm, we now show its efficacy at generating explainable predictions on 5 different vision datasets: ImageNet [37], Places365 [38], CIFAR-{10, 100} [39], and CUB-200 [40]. Specific details about architecture and training protocols used can be found in Appendix §D.2. Our code is available at `https://github.com/r-zip/ip-omp`.

**Semantic dictionaries.** To get a set of semantically relevant concepts, we follow prior work [41], which uses GPT-3 [42] to extract relevant concepts for every class in the dataset by asking prompts like "List the most important features for recognizing something as a class". For more details, we direct the reader to [41]. In this work, we directly use their extracted concepts [43]. For each concept, we obtain their corresponding CLIP embedding by directly using the concept name as input to the text encoder. Some example concepts for Imagenet are shown in col. 2 of Figures 4a and 4b (y-axis).

**Explainable predictions using CLIP-IP-OMP.** We show example runs of the CLIP-IP-OMP algorithm on two images taken from the ImageNet dataset in Figure 4. In particular, given an image, we run IP-OMP on its CLIP embedding for a fixed number of iterations $\tau$, which is the desired explanation length. Then, we make a prediction using the linear classifier, which was trained to learn the mapping between sparse codes of sparsity level $\tau$ (number of non-zero elements) and the class label. All of the CLIP embeddings (text and image) are $\ell_2$-normalized. The signed magnitude of the coefficient in the sparse code indicates how relevant the corresponding atom (text concept) is for the contents of the image. This claim is motivated by CLIP's training loss which incentivizes aligned image-text embedding pairs (that is, the text describes the image contents) to have a high dot product (closer to +1) and unaligned image-text embedding pairs (that is, the text does not correspond to the image contents) to have a small dot product (closer to -1). We also visualize the weights of the trained linear classifier for the predicted class. In Figure 4a, we see that the selected atoms with positive coefficients in the sparse code correspond to characteristics of a tiger like "black and white striped fur" and "prey animal". Similarly, atoms with negative coefficients correspond to concepts not found in a tiger like "salty flavor" and "witch". Although the image is not of a Savannah habitat, we suspect CLIP assigns a positive association between the text embedding for "a Savannah" and the image, since tigers are found in savannah habitats. Interestingly, the sign of the weights the linear classifier assigns to each of the concepts seems to be positively correlated with their sign in the sparse code. Similar observations are made for Figure 4b. Although the concept of "originated in Sussex, England" has a large coefficient in the sparse code, the linear classifier assigns a very low weight to it, indicating a diminished effect of that concept on the prediction "Aircraft Carrier".

**Baseline comparisons.** We compare our method with V-IP and Label-free Concept Bottleneck Models (Lf-CBM) [41], a recently proposed explainable AI algorithm which is similar in spirit to ours. Lf-CBM represents a given image in terms of a concept vector where every dimension

corresponds to a semantic concept and the corresponding feature is the dot product between CLIP's embedding of the image and the text concept. The final prediction is then made by training a sparse linear layer (using elastic net regularization) to map from concept features to labels. Note that their model is sparse in the network weights, whereas ours is sparse in the input features.

For a fair comparison, we use the same set of concepts for all methods. For V-IP, the query is CLIP's text embedding and the answer is its corresponding dot product with the CLIP embedding (since the dot product indicates whether the concept is present in the image). We compare all methods by examining the tradeoff between explanation length and classification accuracy. For V-IP, CLIP-IP-OMP, and Lf-CBM respectively, the explanation lengths are the number of queries selected, the number of dictionary atoms selected, and the number of non-zero

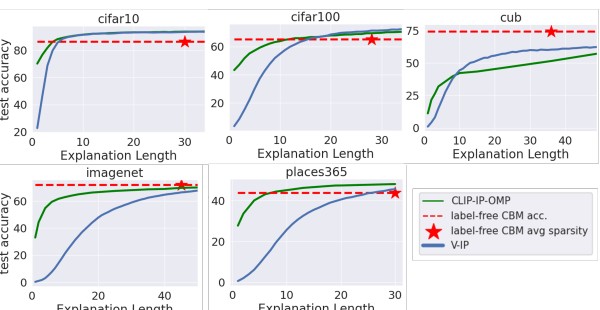

Figure 5: Trade-off between accuracy and explanation length for different methods (best viewed in colour).

weights in the linear classifier for the predicted class. For prediction accuracy, we use V-IP's learned classifier, whereas for our method, we train a separate linear classifier for every sparsity level (the number of iterations of IP-OMP before termination). Unlike CLIP-IP-OMP and V-IP, Lf-CBM does not have any sequential selection strategy. Hence, to compare, we report the average number of non-zero weights per class in the concepts-to-label linear mapping. The results are shown in Figure 5. On most datasets, CLIP-IP-OMP has a better trade-off (greater area-under-the-curve) than V-IP, except CIFAR-100 (where they are competitive) and CUB-200. Notice that for all datasets, in the initial iterations, atoms selected by CLIP-IP-OMP are more predictive for the class (higher test accuracy) than V-IP. This is because the atom selection in IP-OMP explicitly depends on the image content, whereas, the query selection in V-IP (equation 1) solely depends on the answers to the previous queries and the data distribution. On all datasets except CUB and ImageNet (where it is competitive), our method achieves similar accuracy to Lf-CBM, but with a much smaller explanation length (Lf-CBM's explanation length (accuracy) is indicated by the red star (red dashed line)).

**Computational efficiency.** Complementary to being performant at explaining prediction, we would like to stress the simplicity of our method compared to V-IP and Lf-CBM, two contemporary explainable AI methods. Both V-IP and Lf-CBM require costly optimization routines. In V-IP, this is training a querier network to learn what to ask next from data. In Lf-CBM, this is optimizing the sparse linear classifier using the elastic-net regularizer. Finding the right sparsity level (number of non-zero weights in the classifier) in Lf-CBM requires hyperparameter tuning by training multiple models. In comparison, our method uses IP-OMP, which requires no training, to select dictionary atoms. Moreover, the number of iterations of the algorithm controls the explanation lengths.

The main bottleneck of IP-OMP is in extracting the sparse codes since this involves computing projection matrices. For example, our implementation, based on Cholesky decomposition [44], takes $\approx 40$ hours to extract sparse codes of support size 50 (the highest considered in our experiments) for all the 1.28 million images in the ImageNet training set on an NVIDIA RTX A5000 GPU. This will be much faster for smaller support sizes, for example at level 10, it takes $\approx 6$ hours. For comparison, training Lf-CBM [43] on ImageNet for *one set of hyperparameters* takes about 50 hours on the same GPU. Similarly, training the V-IP framework [45] on ImageNet takes about 4 days!

## 5    Conclusions and Limitations

In this work, we show that Orthogonal Matching Pursuit (OMP) is a particular case of Information Pursuit (IP) (modulo a normalization factor) with a suitably chosen set of queries and target variable. We call this IP-derived algorithm for sparse coding IP-OMP. We then present a simple algorithm, called CLIP-IP-OMP, which uses CLIP and IP-OMP to make explainable predictions for visual classification tasks. CLIP-IP-OMP has two limitations. First, its application is restricted to image datasets since it relies on CLIP to encode images and text concepts into a shared space. Second, the atom selection procedure using IP-OMP is independent of the class label; as a result an auxiliary classifier needs to be trained. Future work would aim to address these limitations.

## Acknowledgments and Disclosure of Funding

This research was supported by the Army Research Office under the Multidisciplinary University Research Initiative contract W911NF-17-1-0304, and the NSF–Simons Research Collaboration on the Mathematical and Scientific Foundations of Deep Learning (NSF grant 2031985, Simons grant 814201). Moreover, the authors acknowledge Carolina Pacheco, Liangzu Peng, Jeremías Sulam and Darshan Thaker for their insightful feedback that helped improve the presentation of this work. Finally, we thank Kwan Ho Ryan Chan for helping us with implementing V-IP for baseline comparisons in Section §4.2.

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

# A  Proofs

Throughout, we will write $\Pi_k = \Pi_{D_k}$ to ease the notation.

## A.1  Proof of Lemma 1

*Proof.* Assume that the atoms selected in $D_{\mathcal{L}}$ are linearly independent (since linearly dependent atoms will not be selected by the equation IP-OMP algorithm—more on this in §A.2). The proof relies on Gaussian conditioning.

Notice that since $Z$ is a standard normal distribution and the vector $\zeta = (\langle x, Z \rangle, Z^T D_{\mathcal{L}})^T$ is just a linear transformation of a standard normal, it is also a multivariate normal random variable. Its covariance is

$$\mathrm{cov}(\zeta) = \begin{pmatrix} x^T x & x^T D_{\mathcal{L}} \\ D_{\mathcal{L}}^T x & D_{\mathcal{L}}^T D_{\mathcal{L}} \end{pmatrix}. \tag{12}$$

Now there are two cases.

**Case 1: The covariance matrix has full rank.** In this case,

$$P(\langle x, Z \rangle \mid \{Q = \langle d_Q, z^0 \rangle : Q \in \mathcal{L}\}) = \mathcal{N}(\langle D_{\mathcal{L}} D_{\mathcal{L}}^\dagger x, z^0 \rangle, \|\Pi_{D_{\mathcal{L}}}^\perp x\|_2^2). \tag{13}$$

This result is obtained by observing that the conditioning event $\{Q = \langle d_Q, z^0 \rangle : Q \in \mathcal{L}\}$ is exactly the event $\{D_{\mathcal{L}}^T Z = D_{\mathcal{L}}^T z^0\}$, since $D_{\mathcal{L}}$ consists of atoms selected from $D$ indexed by $\mathcal{L}$. Now, the result is obtained by using formulae for conditioning in multivariate Gaussian random variables [46, equation A.6]. Specifically, the mean is computed as

$$\mathbb{E}[\langle x, Z \rangle \mid D_{\mathcal{L}}^T Z = D_{\mathcal{L}}^T z^0] = x^T D_{\mathcal{L}} (D_{\mathcal{L}}^T D_{\mathcal{L}})^{-1} D_{\mathcal{L}}^T z^0$$
$$= \langle D_{\mathcal{L}} D_{\mathcal{L}}^\dagger x, z^0 \rangle.$$

The variance is computed as

$$\mathrm{var}(\langle x, Z \rangle \mid \{Q = \langle d_Q, z^0 \rangle : Q \in \mathcal{L}\}) = x^T x - x^T D_{\mathcal{L}} (D_{\mathcal{L}}^T D_{\mathcal{L}})^{-1} D_{\mathcal{L}}^T x$$
$$= x^T (I - D_{\mathcal{L}} (D_{\mathcal{L}}^T D_{\mathcal{L}})^{-1} D_{\mathcal{L}}^T) x$$
$$= x^T (I - \Pi_{D_{\mathcal{L}}}) x$$
$$= x^T \Pi_{D_{\mathcal{L}}}^\perp x$$
$$= \|\Pi_{D_{\mathcal{L}}}^\perp x\|_2^2,$$

which is exactly the variance parameter for the normal distribution in equation 13. The result in the Lemma is then just the analytical form of the differential entropy of a Gaussian [47, equation 9.9].

**Case 2: The covariance matrix is singular.** This would be the case when $x$ can be expressed as a linear combination of the columns (atoms) of $D_{\mathcal{L}}$. Intuitively, looking at equation 13, one can conclude that in this case the conditional distribution of $\langle x, Z \rangle$ given the history would be a Dirac delta distribution since the variance is $0$. Consequently, the differential entropy would converge to $-\infty$.

This argument can be made rigorous as follows. Consider a family of random variables defined as $\zeta_\sigma = \langle x, Z \rangle + \sigma \eta$, where $\eta$ is an independent standard normal distribution. We can conclude from Slutsky's theorem [48, Theorem A.14.9] that

$$\zeta_\sigma \xrightarrow{d} \langle x, Z \rangle \text{ as } \sigma \to 0,$$

where $\xrightarrow{d}$ refers to convergence in distribution. Observe that

$$\forall \sigma > 0 \quad P(\zeta_\sigma \mid \{Q = \langle d_Q, z^0 \rangle : Q \in \mathcal{L}\}) = \mathcal{N}(\langle D_{\mathcal{L}} D_{\mathcal{L}}^\dagger x, z^0 \rangle, \sigma^2),$$

since $\|\Pi_{D_{\mathcal{L}}}^\perp x\|_2^2 = 0$ (recall in this case $x$ is a linear combination of columns of $D_{\mathcal{L}}$). This sequence of distributions is known to converge in distribution to the the Dirac delta distribution, thus proving the desired result. $\qquad\square$

## A.2 Proof of Theorem 1

*Proof.* **First iteration.** Assume that $x$ is not one-sparse in $D$ (this case will be covered separately at the end of the proof). For the first iteration ($k = 1$),

$$Q_1 = \underset{Q^j \in \mathcal{Q}}{\arg\max}\, I(Q^j; \langle x, Z \rangle) = \underset{d^j \in D}{\arg\max}\, I(\langle d^j, Z \rangle; \langle x, Z \rangle),$$

where the second equality is just from definition of the queries and their answers in this case.

Now

$$I(\langle d^j, Z \rangle; \langle x, Z \rangle) = h(\langle d^j, Z \rangle) - h(\langle d^j, Z \rangle \mid \langle x, Z \rangle), \tag{14}$$

where $h(\cdot)$ is the differential entropy [47, Definition 9.1].[5] Since $Z$ is Gaussian, $\langle d^j, Z \rangle$ and $\langle x, Z \rangle$ are jointly Gaussian. Therefore we can compute the entropies in equation 14 as [47, Example 9.1.2]

$$h(\langle d^j, Z \rangle) = \frac{1}{2} \log(2\pi e \operatorname{var}(\langle d^j, Z \rangle))$$

and

$$h(\langle d^j, Z \rangle \mid \langle x, Z \rangle) = \frac{1}{2} \log(2\pi e \operatorname{var}(\langle d^j, Z \rangle \mid \langle x, Z \rangle)).$$

Since $\operatorname{cov}(Z) = I_m$ and $\mathbb{E}Z = 0$, we get

$$\operatorname{var}(\langle d^j, Z \rangle) = \mathbb{E}\langle d^j, Z \rangle^2 = \|d^j\|_2^2.$$

Considering the joint covariance

$$\operatorname{cov}\left( \begin{pmatrix} \langle d^j, Z \rangle \\ \langle x, Z \rangle \end{pmatrix} \right) = \begin{pmatrix} \|d^j\|_2^2 & \langle d^j, x \rangle \\ \langle x, d^j \rangle & \|x\|_2^2 \end{pmatrix},$$

and applying a basic identity for Gaussian conditioning [46, Equation A.6], we obtain

$$\operatorname{var}(\langle d^j, Z \rangle \mid \langle x, Z \rangle) = \|d^j\|_2^2 - \frac{\langle d^j, x \rangle^2}{\|x\|_2^2}.$$

Thus for iteration $k = 1$, IP-OMP amounts to solving

$$Q_1 = \underset{d^j \in D}{\arg\max}\, \frac{1}{2} \log\left( \frac{\|d^j\|_2^2}{\|d^j\|_2^2 - \langle d^j, x \rangle^2 / \|x\|_2^2} \right)$$

$$= \underset{d^j \in D}{\arg\max}\, \frac{|\langle d^j, x \rangle|}{\|d^j\|_2 \|x\|_2}. \tag{15}$$

**Iterations between the first and last.** Next consider iterations between the first and last in IP-OMP (the last iteration will be discussed separately). For all $1 \leq k < \tau - 1$, the query selection strategy is as follows:

$$Q_{k+1} = \underset{Q^j \in \mathcal{Q}}{\arg\max}\, I(Q^j; \langle x, Z \rangle \mid \mathcal{H}_{1:k}) = \underset{d^j \in D}{\arg\max}\, I(\langle d^j, Z \rangle; \langle x, Z \rangle \mid \mathcal{H}_{1:k}). \tag{16}$$

As before, we assume that $d^j$, $x$, and the columns of $D_k$ are linearly independent (we will address collinearity issues later). Expanding the Mutual Information (MI) into a difference of differential entropies, we obtain

$$I(\langle d^j, Z \rangle; \langle x, Z \rangle \mid \mathcal{H}_{1:k}) = h(\langle d^j, Z \rangle \mid \mathcal{H}_{1:k}) - h(\langle d^j, Z \rangle \mid \langle x, Z \rangle, \mathcal{H}_{1:k}). \tag{17}$$

Notice that $\langle d^j, Z \rangle$, $\langle x, Z \rangle$, and $D_k^T Z$ are jointly Gaussian. So, similarly to the first iteration, the MI objective simplifies to

$$I(\langle d^j, Z \rangle; \langle x, Z \rangle \mid \mathcal{H}_{1:k}) = \frac{1}{2} \log(2\pi e \operatorname{var}(\langle d^j, Z \rangle \mid \mathcal{H}_{1:k})) - \frac{1}{2} \log(2\pi e \operatorname{var}(\langle d^j, Z \rangle \mid \langle x, Z \rangle, \mathcal{H}_{1:k})).$$

---

[5]If $x$ is one-sparse in $D$, then this expansion will not hold since the joint distribution of $\zeta = (\langle d^j, Z \rangle, \langle x, Z \rangle)^T$ will be degenerate for some $j$. However, the mutual information can still be suitably defined. We defer the details of this case to the end of the proof.

The history can be written $\mathcal{H}_{1:k} = \{D_k^T Z = D_k^T z^0\}$, so we compute the requisite conditional variances from the joint covariance matrix

$$
\mathrm{cov}\left(\begin{pmatrix} \langle d^j, Z \rangle \\ \langle x, Z \rangle \\ D_k^T Z \end{pmatrix}\right) = \begin{pmatrix} \|d^j\|_2^2 & (d^j)^T x & (d^j)^T D_k \\ x^T d^j & \|x\|_2^2 & x^T D_k \\ D_k^T d^j & D_k^T x & D_k^T D_k \end{pmatrix}. \tag{18}
$$

We once again apply the Gaussian conditioning formula, and rewrite the difference of logs as the log of a quotient to obtain

$$
I(\langle d^j, Z \rangle; \langle x, Z \rangle \mid \mathcal{H}_{1:k}) = \frac{1}{2} \log \left( \frac{(d^j)^T \left[ I - D_k (D_k^T D_k)^{-1} D_k^T \right] d^j}{(d^j)^T \left[ I - \begin{bmatrix} x & D_k \end{bmatrix} \begin{pmatrix} x^T x & x^T D_k \\ D_k^T x & D_k^T D_k \end{pmatrix}^{-1} \begin{bmatrix} x & D_k \end{bmatrix}^T \right] d^j} \right). \tag{19}
$$

Define $C_k = [D_k \ \ x]$, let $\Pi_{C_k}$ denote the orthogonal projection onto $\mathrm{range}(C_k)$, and let $\Pi_{C_k}^\perp$ denote the orthogonal projection onto $\mathrm{range}(C_k)^\perp$. We can rewrite equation 19 in terms of projections as

$$
I(\langle d^j, Z \rangle; \langle x, Z \rangle \mid \mathcal{H}_{1:k}) = \frac{1}{2} \log \left( \frac{\langle d^j, \Pi_k^\perp d^j \rangle}{\langle d^j, \Pi_{C_k}^\perp d^j \rangle} \right). \tag{20}
$$

**Lemma 3** ([49], p. 323). *Let $A$ be partitioned as $[A_1 \ A_2]$ with $A_1 \in \mathbb{R}^{m \times p}$ and $\mathrm{rank}(A_1) = p$, $A_2 \in \mathbb{R}^{m \times (n-p)}$, and $\mathrm{rank}(A_2) = n - p$. Let $\Pi_{A_1}$ be the orthogonal projection onto the columns of $A_1$ and $B = \Pi_{A_1}^\perp A_2$. Then the orthogonal projection, $\Pi_A$, onto the columns of $A$ can be written as $\Pi_A = \Pi_{A_1} + B(B^T B)^{-1} B^T$.*

Applying the above lemma to $A = C_k$, we have

$$
\Pi_{C_k} = \Pi_k + \frac{1}{\|\Pi_k^\perp x\|_2^2} \Pi_k^\perp x x^T \Pi_k^\perp,
$$

which, recalling the idempotence of $\Pi_k$, implies

$$
\Pi_{C_k}^\perp = I - \Pi_{C_k}
$$
$$
= \Pi_k^\perp \left( I - \frac{1}{\|\Pi_k^\perp x\|_2^2} x x^T \right) \Pi_k^\perp.
$$

Thus we can simplify the denominator of equation 20 to

$$
\langle d^j, \Pi_{C_k}^\perp d^j \rangle = \left\langle d^j, \ \Pi_k^\perp \left( I - \frac{x x^T}{\|\Pi_k^\perp x\|_2^2} \right) \Pi_k^\perp d^j \right\rangle
$$
$$
= \|\Pi_k^\perp d^j\|_2^2 - \frac{\langle \Pi_k^\perp d^j, x \rangle^2}{\|\Pi_k^\perp x\|_2^2}
$$
$$
= \|\Pi_k^\perp d^j\|_2^2 - \frac{\langle \Pi_k^\perp d^j, \Pi_k^\perp x \rangle^2}{\|\Pi_k^\perp x\|_2^2},
$$

where in the last equation we have used the symmetry and idempotence of $\Pi_k^\perp$. Substituting into equation 20, we obtain

$$
\underset{d^j \in D, \|\Pi_k^\perp d^j\|_2 \neq 0}{\arg\max} I(\langle d^j, Z \rangle; \langle x, Z \rangle \mid \mathcal{H}_{1:k}) = \underset{d^j \in D, \|\Pi_k^\perp d^j\|_2 \neq 0}{\arg\max} \frac{\|\Pi_k^\perp d^j\|_2^2}{\|\Pi_k^\perp d^j\|_2^2 - \langle \Pi_k^\perp d^j, \Pi_k^\perp x \rangle^2 / \|\Pi_k^\perp x\|_2^2}
$$
$$
= \underset{d^j \in D, \|\Pi_k^\perp d^j\|_2 \neq 0}{\arg\max} \frac{1}{1 - \langle \Pi_k^\perp d^j, \Pi_k^\perp x \rangle^2 / (\|\Pi_k^\perp x\|_2^2 \|\Pi_k^\perp d^j\|_2^2)}
$$
$$
= \underset{d^j \in D, \|\Pi_k^\perp d^j\|_2 \neq 0}{\arg\max} \frac{|\langle \Pi_k^\perp d^j, \Pi_k^\perp x \rangle|}{\|\Pi_k^\perp d^j\|_2 \|\Pi_k^\perp x\|_2},
$$
$$
= \underset{d^j \in D}{\arg\max} I(\langle d^j, Z \rangle; \langle x, Z \rangle \mid \mathcal{H}_{1:k}),
$$

which is the required query as in equation 16. The last equality is obtained by observing that if for any $d' \in D$, $\|\Pi_k^\perp d'\|_2 = 0$ then $d'$ can be written as a linear combination of atoms corresponding to queries selected by IP-OMP in the first $k$ iteration. As a result, the random variable $\langle d', Z \rangle$ is a function of the the previous $k$ queries selected, namely $D_k^T Z$. Hence, $\langle d', Z \rangle$ is conditionally independent of $\langle x, Z \rangle$ given history $\mathcal{H}_{1:k} = \{D_k^T Z = D_k^T z^0\}$. Since the mutual information between independent random variables is zero, the fourth equality follows.[6]

**Last iteration.** The last iteration for IP-OMP needs special care for scenarios in which there is no noise and the observed signal $x = D\beta$, for some unknown $\beta \in \mathbb{R}^n$. This implies that if the algorithm is carried out for large enough $\tau - 1$ iterations, then eventually, we would reach that stage such that there exists an atom $d_\tau \in D$, such that $x$ will lie in the span of $[D_{\tau-1} \quad d_\tau]$. In such scenarios, the joint measure of random variables $(\langle x, Z \rangle, D_{\tau-1}^T Z, \langle d_\tau, Z \rangle)$ is not absolutely continuous with respect to the Lesbegue measure, and thus is not defined according to equation 17. However, we can still define it as the limiting conditional mutual information for a family of random variables, $\zeta_\sigma = \langle x, Z \rangle + \sigma \eta$, (with $\eta \sim \mathcal{N}(0, 1)$ independent of $Z$) with $\langle d_\tau, Z \rangle$ in the limit $\sigma \to 0$. Specifically, for any $\sigma > 0$, we have

$$I(\langle d_\tau, Z \rangle; \zeta_\sigma \mid \mathcal{H}_{1:\tau-1}) = I(\langle d_\tau, Z \rangle; \zeta_\sigma \mid D_{\tau-1}^T Z = D_{\tau-1}^T z^0) \tag{21}$$
$$= h(\zeta_\sigma \mid D_{\tau-1}^T Z = D_{\tau-1}^T z^0) - h(\zeta_\sigma \mid D_{\tau-1}^T Z = D_{\tau-1}^T z^0, \langle d_\tau, Z \rangle).$$

Since $\zeta_\sigma$ is jointly Gaussian with the random variables, $D_{\tau-1}^T Z$ and $\langle d_\tau, Z \rangle$, equation 21 can be rewritten as, for any $\sigma > 0$,

$$I(\langle d_\tau, Z \rangle; \zeta_\sigma \mid \mathcal{H}_{1:\tau-1}) = \frac{1}{2} \log \left( \frac{\mathrm{var}(\zeta_\sigma \mid D_{\tau-1}^T Z = D_{\tau-1}^T z^0)}{\mathrm{var}(\zeta_\sigma \mid D_{\tau-1}^T Z = D_{\tau-1}^T z^0, \langle d_\tau, Z \rangle)} \right)$$
$$= \frac{1}{2} \log \left( \frac{\|\Pi_{D_{\tau-1}}^\perp x\|_2^2 + \sigma^2}{\sigma^2} \right).$$

The second equality is obtained by applying known formulae of Gaussian conditioning as done in equation 19. Note that $\|\Pi_{D_{\tau-1}}^\perp\|_2$ can also be written as $\|\Pi_{\tau-1}^\perp\|_2$ following our convention.

Finally, taking the limit $\sigma \to 0$, we obtain

$$I(\langle d_\tau, Z \rangle; \langle x, Z \rangle \mid \mathcal{H}_{1:\tau-1}) := \lim_{\sigma \to 0} I(\langle d_\tau, Z \rangle; \zeta_\sigma \mid \mathcal{H}_{1:\tau-1}) = \infty,$$

Moreover, notice that since, by assumption, $x$ now lies in the span of $[D_{\tau-1} \quad d_\tau]$, we can express it as $x = [D_{\tau-1} \quad d_\tau]\alpha$, for some $\alpha \in \mathbb{R}^\tau$. This implies that

$$\Pi_{\tau-1}^\perp x = \Pi_{\tau-1}^\perp [D_{\tau-1} \quad d_\tau]\alpha = c \Pi_{\tau-1}^\perp d_\tau$$

for some scalar $c \in \mathbb{R}$. Thus, $\Pi_{\tau-1}^\perp d_\tau$ and $\Pi_{\tau-1}^\perp x$ are collinear vectors indicating that $d_\tau$ achieves the $\arg\max$ to the IP-OMP objective. That is,

$$\max_{d^j \in D} \frac{|\langle \Pi_{\tau-1}^\perp d^j, \Pi_{\tau-1}^\perp x \rangle|}{\|\Pi_{\tau-1}^\perp d^j\|_2 \|\Pi_{\tau-1}^\perp x\|_2} = \frac{|\langle \Pi_{\tau-1}^\perp d_\tau, \Pi_{\tau-1}^\perp x \rangle|}{\|\Pi_{\tau-1}^\perp d_\tau\|_2 \|\Pi_{\tau-1}^\perp x\|_2} = 1.$$

To conclude, we have shown that any atom that results in $x$ being in the span of that atom plus that atoms already selected so far in the first $\tau - 1$ iterations would be the $\arg\max$ of the objective in IP-OMP in the last iteration. This iteration would be the last since all subsequent iterations would have no informative queries left as the conditional mutual information for all remaining queries would be 0, since they would be rendered conditionally independent of the target variable given the history $H_{1:\tau}$ (as described in footnote 6). $\qquad \square$

---

[6] Note that although in this case, the conventional definition of MI is not defined since we do not have a joint density with respect to the Lesbegue measure, we can use the more general defintion of mutual information [50, Definition 2.1] which depends on the log of the Radon-Nikodym derivative of the joint measure with respect to to the product measure (of the marginals) which is always defined. Since $\langle d', Z \rangle$ is independent of $\langle x, Z \rangle$ given the history this Radon-Nikodym derivative would be the constant function 1 and hence MI would be zero.

### A.3 Proof of equation 7

*Proof.* Recall that $\mathcal{H}_{1:\tau} = \{D_\tau^T Z = D_\tau^T z^0\}$. As discussed in equation 12, the joint covariance

$$\text{cov}\left(\begin{pmatrix} \langle x, Z \rangle \\ D_\tau^T Z \end{pmatrix}\right) = \begin{pmatrix} x^T x & x^T D_\tau \\ D_\tau^T x & D_\tau^T D_\tau \end{pmatrix}, \tag{22}$$

where $D_{\mathcal{L}}$ is replaced by $D_\tau$ since we are no longer dealing with an arbitrary subset of atoms (denoted by $\mathcal{L}$) but rather with the $\tau$ atoms selected in the first $\tau$ iterations of IP-OMP.

Consequently from equation 13 we conclude that

$$P(\langle x, Z \rangle \mid \mathcal{H}_{1:\tau}) = \mathcal{N}(\langle D_\tau D_\tau^\dagger x, z^0 \rangle, \|\Pi_{D_\tau}^\perp x\|_2^2).$$

The result in equation 7 is then obtained by the fact that the mode of a Gaussian is just its mean.

We conclude this proof with a note that in the case when $x$ can be expressed as a linear combination of atoms in $D_\tau$, the distribution $P(\langle x, Z \rangle \mid \mathcal{H}_{1:\tau})$ is a Dirac delta distribution with all its mass concentrated on $\langle D_\tau D_\tau^\dagger x, z^0 \rangle$ (see Case 2 in §A.1). As a result, even in this case, IP-OMP's prediction of the target would be $\langle D_\tau D_\tau^\dagger x, z^0 \rangle$. □

### A.4 Proof of Lemma 2

*Proof.* Notice from the query selection equations for IP-OMP that the choice of the next query is independent of the particular realization of $Z$. This means that, given $x$, for all realizations of $Z \in \mathbb{R}^m$ ($\mathbb{R}^m$ is the support of an $m$-dimensional standard normal), IP-OMP will select the exact same sequence of $\tau$ dictionary atoms. Thus, using the result from equation 7 we can conclude that IP-OMP's prediction of the target for any $z \in \mathbb{R}^m$ will be $\langle D_\tau D_\tau^\dagger x, z \rangle$. □

## B  Sparse Recovery Experiments: Details and Further Results

### B.1  General Setup

To test the sparse recovery performance of IP-OMP against that of OMP, we experimented with IP-OMP on synthetic data. Our setup is a modified version of Tropp and Gilbert's [24], who studied recovery guarantees for OMP with Gaussian dictionaries. For $n = 256$ and $n = 1024$, we generated random dictionaries $D_o$ and sparse codes $\beta_o$ for varying numbers of measurements $m$ and sparsity levels $s$. Each column of the dictionary $D_o$ was drawn from the uniform distribution on the unit sphere in $\mathbb{R}^m$.[7] The sparse codes $\beta_o$ were generated by fixing some sparsity level $s$ and drawing the support $S$ from all cardinality-$s$ subsets of $\{1, \ldots, n\}$ uniformly at random. On the support ($i \in S$), the coefficients $\beta_o'$ were generated according to two schemes:

   (i) $\beta_o \sim \mathcal{N}(0, 1)$ (Gaussian nonzero coefficients), and

   (ii) $\beta_o = 1$ (constant nonzero coefficients).

For $i \notin S$, the corresponding coefficients were set to zero for both of the above schemes. The second setting, which is supposed to be more challenging for OMP, matches the generation of sparse coefficient vectors in [24].

For each setting of $n$ and nonzero coefficients (Gaussian or constant), we report recovery for a different grid of $m$ values. These values, given in Table 1, were selected to emphasize the transition region in the recovery probability curves. For both small ($n = 256$) settings, we simulated sparsities $s \in \{4, 10, 16, \ldots, 36\}$. For both large ($n = 1024$) settings, we simulated sparsities $s \in \{4, 6, 8, \ldots, 16\}$.

This setup was used to test the sparse recovery of IP-OMP and OMP in both noiseless and noisy settings. Note that for noisy settings, we only experimented with the Gaussian nonzero coefficients for generating $\beta_o$ on the sampled support (setting (i)) due to resource limitations. We will first elaborate on the noiseless setup and then the noisy case.

---

[7] In high dimensions, Gaussian vectors concentrate in norm [51, Chapter 3], which justifies applying Tropp and Gilbert's problem settings to spherical dictionaries.

Table 1: Values of $m$ in sparse recovery experiments.

| Nonzero coefficients | $n = 256$ | $n = 1024$ |
|---|---|---|
| Gaussian | $\{4, 16, 28, \ldots, 208\}$ | $\{5, 10, 15, \ldots, 115\}$ |
| Constant | $\{4, 16, 28, \ldots, 256\}$ | $\{5, 10, 15, \ldots, 205\}$ |

## B.2 Experiments without Noise

For each $(m, n, s)$ triple, we generated 1000 dictionary-code-signal triples $(D_o, \beta_o, x_o)$ as described in Appendix §B.1 and ran both OMP and IP-OMP for $\tau = s$ iterations. For each trial, we checked whether OMP and IP-OMP recover $\beta_o$ exactly. Writing $\hat{\beta}_{\text{Alg.}}$ to denote the IP or IP-OMP estimate of $\beta_o$, we define exact recovery to be $\text{NMSE}(\hat{\beta}_{\text{Alg.}}, \beta_o) < 10^{-14}$.[8] For Gaussian nonzero coefficients, we report the fraction of successful trials for each algorithm and $(m, n, s)$ triple in Figure 2. For constant nonzero coefficients, we report the same in Figure 6. Both Figure 2 and Figure 6 were inspired by similar figures in [24].

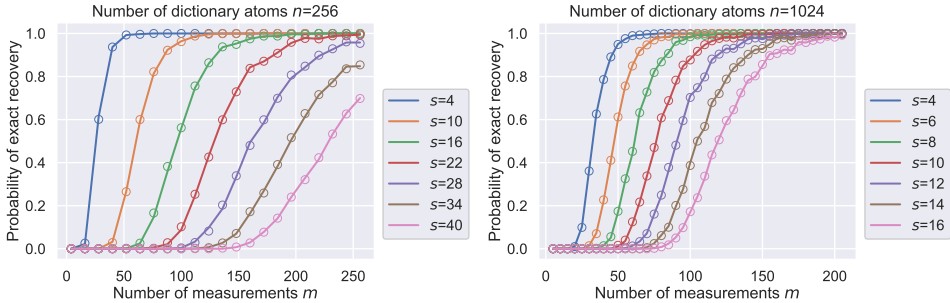

Figure 6: Exact recovery probabilities for OMP and IP-OMP with constant nonzero coefficients. Each solid line represents the recovery probability curve for OMP at a fixed sparsity level, as the number of measurements $m$ increases. The overlaid circles are the corresponding probabilities of recovery by IP-OMP. The left panel is the small setting ($n = 256$), while the right is the large setting ($n = 1024$).

Qualitatively, the constant nonzero coefficient results in Figure 6 are similar to the Gaussian nonzero coefficient results in Figure 2. As discussed in §3, the estimates $\hat{\beta}_{\text{Alg.}}$ degrade gracefully as the measurement rate $m/n$ (a measure of the indeterminacy of the problem) tends to zero and sparsity rate $s/m$ (a measure of the complexity of the solution) tends to one. However, the constant nonzero coefficient case is more challenging to recover (probability curves for each sparsity $s$ are shifted to the right relative to the Gaussian nonzero coefficient case). The difficulty of the constant nonzero coefficient setting has been noted previously for OMP [24].

## B.3 Experiments with Noise

As described in §3, we extended the setup in Appendix §B.1 with the addition of Gaussian noise. That is, for each draw of the uniform spherical dictionary $D_o$ and sparse code $\beta_o \sim \mathcal{N}(0, I)$, we also sampled noise $e \sim \mathcal{N}(0, \sigma^2 I)$ independent of $D_o$ and $\beta_o$. The variance $\sigma^2$ of this noise was set to achieve a particular $\mathbb{E}[\text{SNR}]$ according to equation 23. This was done to ensure that the level of noise added to observed signal is consistent across different settings of $(m, n, s)$ as described in Table 1. The natural scale for varying the $\mathbb{E}[\text{SNR}]$ values is logarithmic; in particular, we chose to use the decibel (dB) scale, where we take $\mathbb{E}[\text{SNR}]$ in dB units to be defined as

$$\mathbb{E}[\text{SNR}] = 10 \log_{10} \mathbb{E}\left[\frac{\|D_o \beta_o\|_2^2}{\|e\|_2^2}\right] \text{ dB}.$$

Expected SNR values were swept from 5 dB to 20 dB in 5 dB intervals. Thus, for each $(m, n, s, \mathbb{E}[\text{SNR}])$ setting, we generated 1000 $(D_o, \beta_o, x_o, e)$ tuples and ran OMP and IP-OMP

---

[8]Recall that the NMSE was defined in equation 9.

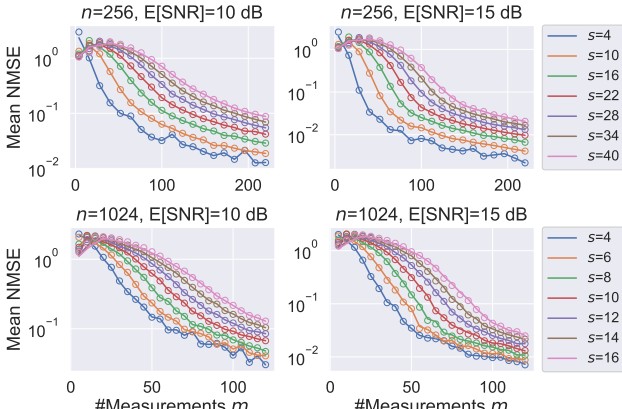

Figure 7: Further numerical results on sparse recovery where the signal is corrupted by noise. These result add to those in Figure 3. On the $y$-axis we report $\mathrm{NMSE}(\hat{\beta}_{\mathrm{Alg.}}, \beta)$ for OMP (lines) and IP-OMP (overlaid circles). On the $x$-axis, we report the number of measurements (signal dimension) $m$. Each value of the sparsity $s$ corresponds to a different color.

for $s$ iterations. Finally, for each trial, we measured the quality of the estimates $\hat{\beta}_{\mathrm{Alg.}}$ by their NMSE as defined in equation 9. The results for $\mathbb{E}[\mathrm{SNR}] \in \{5\,\mathrm{dB}, 20\,\mathrm{dB}\}$ are shown in Figure 3 in the main paper and the results for $\mathbb{E}[\mathrm{SNR}] \in \{10\,\mathrm{dB}, 15\,\mathrm{dB}\}$ are reported in Figure 7.

Now we present a proposition that gives a formula for $\mathbb{E}[\mathrm{SNR}]$ in terms $\sigma, m$ and $s$ when $D_o, \beta_o$, and $e$ are distributed according to our above described sampling process. For technical consistency, we denote $\mathcal{D}, \mathcal{B}, E$ as the random variables and $D_0, \beta_0$ and $e$ as their respective realizations.

**Proposition 1.** *Let $\mathcal{D}$ be a $m \times n$-dimensional random matrix with columns distributed independently and uniformly on the unit sphere. Let $\mathcal{S}$ be a random support of size $s$, chosen uniformly from all size-$s$ subsets of $\{1, 2, \ldots, n\}$. Let $\mathcal{B}$ be a random $s$-sparse vector with support on $\mathcal{S}$ and with coefficients zero-mean, unit-variance Gaussian as described in § B. Let $E \sim \mathcal{N}(0, \sigma^2 I)$. Then for the noisy measurement model $\widetilde{X} = \mathcal{D}\mathcal{B} + E$, we have*

$$\mathbb{E}[\mathrm{SNR}] := \mathbb{E}_{\mathcal{D}, \mathcal{B}, E}\left[\frac{\|\mathcal{D}\mathcal{B}\|_2^2}{\|E\|_2^2}\right] = \frac{s}{\sigma^2(m-2)}, \tag{23}$$

*or on dB scale,*

$$\mathbb{E}[\mathrm{SNR}] = 10\log_{10}\left(\frac{s}{\sigma^2(m-2)}\right) \mathrm{dB}.$$

*Proof.* Since $E$ is independent of $\mathcal{D}$ and $\mathcal{B}$, we can split the expectation in the definition of the expected SNR into

$$\mathbb{E}[\mathrm{SNR}] = \mathbb{E}\left[\|\mathcal{D}\mathcal{B}\|_2^2\right] \mathbb{E}\left[\frac{1}{\|E\|_2^2}\right].$$

We will first compute $\mathbb{E}[\|\mathcal{D}\mathcal{B}\|_2^2]$:

$$\begin{aligned}
\mathbb{E}\|\mathcal{D}\mathcal{B}\|_2^2 &= \mathbb{E}\left[\mathcal{B}^T \mathcal{D}^T \mathcal{D}\mathcal{B}\right] \\
&= \mathbb{E}\,\mathrm{tr}(\mathcal{B}^T \mathcal{D}^T \mathcal{D}\mathcal{B}) \\
&= \mathbb{E}\,\mathrm{tr}(\mathcal{B}\mathcal{B}^T \mathcal{D}^T \mathcal{D}) \\
&= \mathrm{tr}\left(\mathbb{E}[\mathcal{B}\mathcal{B}^T]\mathbb{E}[\mathcal{D}^T \mathcal{D}]\right).
\end{aligned} \tag{24}$$

Rewriting as an inner product, we have

$$\mathbb{E}[\mathcal{D}^T \mathcal{D}]_{ij} = \mathbb{E}\left[\langle \mathcal{D}^i, \mathcal{D}^j \rangle\right] = \delta_{ij},$$

where $\mathcal{D}^i$ is the $i^{\mathrm{th}}$ column of random matrix $\mathcal{D}$ and $\delta_{ij}$ is the Kronecker delta. The last equality if obtained by observing that since $\|\mathcal{D}^j\|_2^2$ is a constant one and $\mathcal{D}^i, \mathcal{D}^j$ are independent for $i \neq j$. We thus conclude that $\mathbb{E}[\mathcal{D}^T \mathcal{D}] = I$.

Notice that $\mathcal{B}$ depends on the random support $\mathcal{S}$. Furthermore, since $\mathbb{E}[\mathcal{D}^T \mathcal{D}] = I$ and

$$\mathrm{tr}(\mathbb{E}[\mathcal{B}\mathcal{B}^T]\mathbb{E}[\mathcal{D}^T \mathcal{D}]) = \mathrm{tr}(\mathbb{E}[\mathcal{B}\mathcal{B}^T]),$$

it suffices to consider the diagonal entries of $\mathbb{E}[\mathcal{B}\mathcal{B}^T]$:

$$\mathbb{E}[\mathcal{B}\mathcal{B}^T]_{ii} = \mathbb{E}\Big[\mathbb{E}\big[\mathcal{B}\mathcal{B}^T \mid \mathcal{S}\big]\Big]_{ii}$$
$$= \mathbb{E}\big[\mathbb{E}[\mathcal{B}_i^2 \mid \mathcal{S}]\big]$$
$$= \mathbb{E}[\mathbf{1}_{\mathcal{S}}(i)], \qquad\qquad \mathbf{1}_{\mathcal{S}}(i) \text{ is the set indicator function for random set } \mathcal{S}$$
$$= \mathbb{P}[i \in \mathcal{S}],$$

where the third equality results from the fact that coefficients $\mathcal{B}_i$ on the support (i.e., $i \in \mathcal{S}$) are Gaussian with zero mean and unit variance. In the fourth equality $\mathbb{P}[i \in \mathcal{S}]$ is the probability that a randomly drawn subset $\mathcal{S}$ would include element $i$ in it. To compute $\mathbb{P}[i \in \mathcal{S}]$, we count the number of cardinality-$s$ subsets of $\{1, 2, ..., n\}$ containing $i$ and divide by the total number of cardinality-$s$ subsets. This is described in detail in the following equations where we use $S$ to denote the different values $\mathcal{S}$ takes on the sample space.

$$\mathbb{P}[i \in \mathcal{S}] = \frac{|\{S \in 2^{[n]} : |S| = s,\ i \in S\}|}{|\{S \in 2^{[n]} : |S| = s\}|}$$
$$= \frac{\binom{n-1}{s-1}}{\binom{n}{s}}$$
$$= \frac{(n-1)!}{(s-1)!(n-1-s+1)!} \frac{s!(n-s)!}{n!}$$
$$= \frac{(n-1)!}{(s-1)!(n-s)!} \frac{s(s-1)!(n-s)!}{n(n-1)!}$$
$$= \frac{s}{n}.$$

Thus we have

$$\mathbb{E}[\mathcal{B}\mathcal{B}^T]_{ii} = \frac{s}{n}, \quad i = 1, \ldots, n.$$

Conclude that

$$\mathrm{tr}(\mathbb{E}[\mathcal{B}\mathcal{B}^T]\mathbb{E}[\mathcal{D}^T\mathcal{D}]) = \mathrm{tr}(\mathbb{E}[\mathcal{B}\mathcal{B}^T])$$
$$= \sum_{i=1}^{n} \frac{s}{n}$$
$$= s. \tag{25}$$

Rewriting $\|E\|_2^2$ as

$$\|E\|_2^2 = \sum_{i=1}^{m} E_i^2,$$

and recalling that $E_i \overset{\text{i.i.d.}}{\sim} \mathcal{N}(0, \sigma^2)$, we see that $(1/\sigma^2)\|E\|_2^2$ has a $\chi^2$ distribution with $m$ degrees of freedom. It follows that $\sigma^2/\|E\|_2^2$ has an inverse-$\chi^2$ distribution with $m$ degrees of freedom, which has an expected value of $1/(m-2)$ [52, pp. 578–579, 583]. Thus

$$\mathbb{E}\left[\frac{1}{\|E\|_2^2}\right] = \frac{1}{\sigma^2(m-2)}.$$

Finally, we plug the value for $\mathrm{tr}(\mathbb{E}[\mathcal{B}\mathcal{B}^T]\mathbb{E}[\mathcal{D}^T\mathcal{D}])$ from equation 25 into equation 24 and multiply by $\mathbb{E}[1/\|E\|_2^2]$:

$$\mathbb{E}[\mathrm{SNR}] = \mathrm{tr}\left(\mathbb{E}\left[\mathcal{B}\mathcal{B}^T\right]\mathbb{E}\left[\mathcal{D}^T\mathcal{D}\right]\right)\mathbb{E}\left[\frac{1}{\|E\|_2^2}\right]$$
$$= \frac{s}{\sigma^2(m-2)}. \qquad\qquad \square$$

# C  Prior Work on Explainable AI

In this section we review prior work on explainable AI. For the references herein please refer to the References section at the end of this Appendix.

Modern deep networks are often criticized for their opaque nature resulting in a lack of transparency in their decision-making process [53]. As a result, they are often perceived as "black-boxes". Initial attempts to explain deep networks relied on post-hoc techniques like sensitivity-based or gradient-based analyses of the inputs and the outputs of a trained network [54, 55, 56, 57, 58, 59]. These methods attempt to explain a deep network's decision by generating a saliency map reflecting which parts of the input were most important for a particular prediction. The veracity of these methods at explaining the true underlying decision-making process has been questioned in several recent works [60, 61, 62, 63, 64, 65]. Moreover, the explanations obtained, in terms of importance of raw input features (like pixels) to the predicted output, are often not meaningful explanations to stakeholders [5, 41]. As a result, recent works have advocated that machine learning (ML) models should be explainable by design [66, 5, 67].

Over the past few years, there has been a flurry of papers that propose ML algorithms that are explainable by design; however, they vary in their definitions of what it means to be explainable. These can be broadly classified into three categories:

1. *Regularization to make deep networks more interpretable.* These methods propose to regularize the training of a deep network such that their decisions can be well-approximated locally by an interpretable ML model like linear classifiers [68, 69] or decision trees [70]. However, as in post-hoc techniques to explainability, the effect of these approximations on the fidelity of the generated explanations to the deep networks decision-making process is unknown. Alternatively, Wong et al. [71] show that learning a sparse final layer of the network improves interpretability, but requires post-hoc methods like LIME to explain the deep features used by the sparse layer.

2. *Explanations based on learnt concepts.* These methods propose to jointly learn prototypes or latent vectors from data and use these latent vectors to make a prediction [72, 73, 74, 75, 76]. The final prediction is then explained in terms of these learnt concepts, whose interpretation is based on exemplar-based reasoning. For example, Sarkar et al. [72] look at images that maximally activate a particular feature of the latent vector to decide what that feature represents. Similarly, Li et al. [75] replace their learnt prototype by the embedding of the nearest-neighbour training image patch. This selected patch then serves as an interpretation of the prototype. A limitation of these methods is that it is not necessary for these learnt prototypes/latents to be interpretable to the end-user [5].

3. *Explanations based on user-defined concepts.* In these methods, the user first defines a set of concepts that are relevant for the prediction task, along with annotated datasets indicating the presence or absence of concepts in each input. The final prediction is the made based on these concepts, which serve as an explanation for the prediction [5, 15, 67]. In contrast to the previous set of methods, joint training of concepts and label classifiers in these methods is done in a completely supervised manner (the concepts are not latent vectors). In another direction, Chen et al. [77] propose to make layers of a deep network more interpretable by learning a rotation matrix that rotates any selected deep network's layer activations such that every activation (after rotation) corresponds to one of the user-defined concepts. Unlike the previous two categories, the methods in this category have the flexibility of allowing the user to decide on the nature of explanations required for the prediction task by selecting the set of concepts as input to the learning algorithm. Our proposed algorithm CLIP-IP-OMP also belongs to this class of methods.

Despite its advantage, a major limitation of the third category of methods is the requirement for data annotated with these user-defined concepts. This bottleneck was addressed (for image classification tasks) very recently in [41, 78, 79] who showed that CLIP, a large vision-language pre-trained model can be used to provide zero-shot annotations for user-defined concepts. In particular, text embeddings of the concepts are obtained using CLIP's text encoder. The image annotations (whether or not a concept is present or not in the image) is then provided via the dot product between the text embedding of the concept and the image embedding (obtained by passing the image through CLIP's image encoder). This motivated our algorithm CLIP-IP-OMP, which uses CLIP embeddings to construct

the interpretable dictionary. As argued in the main paper, our proposed algorithm is computationally much simpler than existing state-of-the-art methods of explainable-AI (comparing with methods from the third category, which include Variational-IP [15] and Label-free CBM [41]). CLIP has also been applied for semantic interpretation of neurons in a Convolutional Neural Network [80].

# D    CLIP-IP-OMP Training Details

## D.1    Semantic Dictionaries

In principle, a user can define a set of semantic concepts that are relevant for the classification task. A dictionary can then be constructed by obtained CLIP's text embedding for each one of those concepts. In this work, to illustrate the usefullness of our framework, we used the set of concepts used in [41] for each dataset. For completeness, we describe the process in which these concepts were obtained in more detail.

The authors of [41] employed GPT-3 [42], a large language model, to extract an initial set of semantic concpets. They used three prompts for this purpose.

- List the most important features for recognizing something as a "class".
- List the things most commonly seen around a "class".
- Give superclass for the word "class".

In the above prompts "class" is a placeholder that is replaced by the class names present in the dataset. This initial list of concepts was then further pruned to remove redundancies. The authors employed the following set of rules for pruning.

- Any concepts with textual description longer than 30 characters was removed.
- Any concept too similar to the target class was removed. This was done to prevent explanations of a class to be the class itself. To do this, the authors measured the cosine similarity between the concepts and the target classes in the text embedding space and pruned any concept with similarity higher than $0.85$. The cosine similarity was measured by a weighted combination of CLIP's ViT-B/16 text encoder and the all-mpnet-base-v2 sentence encoder.
- Concepts similar to each other in CLIP's text embedding space were pruned to remove duplicate concepts. Specifically, any concept with more than $0.9$ cosine similarity (same weighted combination as above) with any concept already present in their list.
- Concepts not present in the training data were pruned. This was done by removing any concept whose average top-5 CLIP activations (measured as similarity between the concept's CLIP text embedding and the image's CLIP image embedding) was below a pre-defined threshold.
- Finally, the authors employ a projection layer to learn to estimate CLIP's similarity between a text concept and an image given the image's deep features obtained from some Backbone network like Resnet-50 [81] (so at test time CLIP would not be needed). Any concept that this projection layer is not able to estimate well (using some measure of accuracy) is removed.

The number of concepts finally used for each dataset are as follows: 128 for CIFAR-10, 824 for CIFAR-100, 208 for CUB-200, 4523 for ImageNet, and 2207 for Places365.

## D.2    More Training Details for the CLIP-IP-OMP Algorithm

For all our experiments we use the CLIP Vit-B/16 model. In our experimental setup in § 4.2, for a given dataset, we first set the desired number of iterations[9] to run the IP-OMP algorithm for sparse-coding the image embedding in terms of dictionary atoms. This is the desired explanation length. We then obtain sparse-codes for every image in the training set using the IP-OMP algorithm. This creates

---

[9]While we employed this termination criterion in our experiments, one can also alternatively use a variable number of iterations per image by using the entropy-based stopping criterion where one terminates once the norm of the residual is sufficiently small.

a new training dataset $\mathscr{D}'$ where every (image, label) pair is transformed into a (sparse-code, label) pair. A linear network is then trained using the cross-entropy loss using $\mathscr{D}'$. For all datasets we used the same training configuration. We employed SGD with momentum optimizer with learning rate 1.0 and momentum 0.9. To prevent overfitting, we employed a drop-out layer before the linear layer with parameter 0.5. Finally, the linear classifier was trained for 1000 epochs. The input and output neurons for the linear layer for every dataset was taken as the number of concepts in the dictionary and the number of output classes respectively. The prediction probability of the trained classifier is obtained via a softmax on the output.

To obtain the curves in Figure 5 we repeated the above described process for different values of $\tau$. Since, our classifier is linear, training via SGD is relatively fast. The longest time being on the Imagenet and Places365 small datasets (since they have 1.28M and 1.8M images respectively) where every epoch takes about 17 and 50 seconds respectively on a single NVIDIA RTX A5000 GPU with a batch size of 1024.

## E   Example Runs for the CLIP-IP-OMP Algorithm

We show example runs of the CLIP-IP-OMP algorithm on 5 images selected from the test/validation set[10] of the different datasets on which we tested our algorithm. For each dataset, we run the IP-OMP algorithm for a fixed number of iterations $\tau$ and make predictions using a linear classifier. In practice, a user can look at the accuracy vs. explanation length trade-off curves (Figure 5) and decide on a value for $\tau$ based on their requirement (more accuracy vs. ease of interpretation of the results). In this appendix, for every dataset, we select some value for $\tau$ to illustrate the efficacy of the proposed framework at making explainable predictions.

### E.1   ImageNet

For this dataset, we set $\tau = 9$, that is, the sparse code $\hat{\beta}_{\text{IP-OMP}}$ for every image has support[11] of size 9. A linear classifier trained on these sparse codes gets about 63% accuracy, which is within 9 points of what can be achieved if we run the IP-OMP algorithm long enough. This can be seen from Figure 5, where the test accuracy vs. explanation length curve for ImageNet saturates at about 70% accuracy for relatively large $\tau$ (say $\tau = 50$). Results in Figure 9.

1.  In Figure 9a, our model correctly predicts "Great White Shark". The sign of the weights of the linear classifier seem to correlate with the sign of the atom coefficients in the sparse code. The prediction can be attributed to positive concepts like "shark", "a large, white, rounded head", "powerful jaws", and "marine animal", all of which are salient features of a great white shark. Similarly, negative concepts like "swan", "planets", "streetcar", and "a toad" are concepts whose presence would indicate that the image is definitely not of a great white shark.

2.  In Figure 9b, our model correctly predicts "Snow Leopard". The sign of the weights of the linear classifier seem to correlate with the sign of the atom coefficients in the sparse code. The prediction can be attributed to positive concepts like "spotted or striped fur", "a mottled gray and white coat", and "a cub"; which are also the concepts to which the classifier assigns the highest weights. Note other concepts like "a cell" (probably referencing a zoo cell) and "a fence" are also present in the image, and are attributed as positive concepts in the sparse code. However, since these are not salient for the class "Snow Leopard", these are attributed smaller weights by the linear classifier. Thus, our method shows that the model is making the right prediction for the right reasons, something that is very difficult to ascertain in black-box deep networks. Similarly, absence of concepts like "a striped coat" and "a white throat" probably help the classifier to distinguish a snow leopard from other big cats.

3.  In Figure 9c, our model correctly predicts "Goldfish". The sign of the weights of the linear classifier seem to largely correlate with the sign of the atom coefficients in the sparse code. The prediction can be attributed to positive concepts like "a fish tank", "a [sic] aquarium",

---

[10]We use validation sets only for the ImageNet and the Places365 datasets since for these datasets we only have access to the validation set and not the test set.

[11]The support of a vector is the set of indices of all non-zero elements in the vector.

"orange and white stripes" (which is very salient to a goldfish); which are also the concepts on which the classifier assigns the highest weights. Note that "often has a decorative design" is also positively attributed in the sparse code by IP-OMP, most likely because goldfish is often seen as a decorative item in one's household. Indeed, inspecting the GPT3 concepts recovered for the "GoldFish" class in [43], we see that "decoration" is one of the concepts that is supplied by GPT3 for this class. Similarly, negative concepts like "trash", "gym clothes" and "spider" identified by IP-OMP are concepts whose presence would strongly indicate that the object in the image is not a goldfish. Accordingly the linear classifier assigns negative weights to these concepts.

4. In Figure 9d, our model incorrectly predicts "Electric Guitar". The true class is "Acoustic Guitar". Inspecting the selected atoms and the weights of the classifier, it is clear that from the positive concepts—"a guitar case", "a small, handheld instrument", "music", and "equipment"; along with the negative concepts, one cannot distinguish between an electric and an acoustic guitar. Indeed, the second most probable class in this case is "Acoustic Guitar". Note that although the image does not explicitly show "a guitar case", it is a concept positively correlated with the object in the image and CLIP's latent space is such that text embeddings that are related to image content have a high correlation with the image's embedding. Indeed, "a guitar case" is one of the concepts supplied by GPT3 for the class "Acoustic Guitar".

5. In Figure 9e, our model incorrectly predicts "Volcano". We see that the two most salient concepts according to the classifier (atoms to which the classifier assigns highest weight) are "eruption" and "a cone-shaped mountain", which are concepts a human would use to describe a volcano. Other positive concepts recovered by IP-OMP include "aflame", "a white or light-colored body" (probably looking at the bright flame) and "made of lava and ash" (which is the concept attributed the third highest weight by the linear classifier).

## E.2 Places365

For this dataset, we set $\tau = 11$, that is, the sparse code $\hat{\beta}_{\text{IP-OMP}}$ for every image has support of size 11. A linear classifier trained on these sparse codes gets about $45.5\%$ accuracy, which is within 3 points of what can be achieved if we run the IP-OMP algorithm long enough. This can be seen from Figure 5, where the test accuracy vs. explanation length curve for Places365 saturates at about $48\%$ accuracy for relatively large $\tau$ (say $\tau = 30$). Results in Figure 10.

1. In Figure 10a, our model correctly predicts "Crosswalk". We see that the two concepts assigned highest positive value in the sparse code by IP-OMP are "divided by lines or curbs" and "usually found near an airport". However the latter is not really discriminative for a crosswalk since they are found everywhere, and not necessarily at an airport. This is correctly reflected in the weights of the linear classifier, which assign a much higher weight to the concept "divided by lines or curbs" but a relatively low (the lowest positive weight out of all the 11 concepts in the sparse code) to "usually found near an airport". Negative concepts like "extensive grounds and gardens", "a tile floor"', "often found on barns", and "found floating in the ocean" are strong indicators that the scene in the image is not of a crosswalk.

2. In Figure 10b, our model correctly predicts "Wet Bar". Interestingly, the two most important concepts (largest classifier weights) are "cabinets and counter-tops" and "a dishwashing area". The main distinguishing feature between a dry bar and a wet bar is that a wet bar has a washing sink. Other positively identified concepts by IP-OMP are "brochures", "resort"' and "a protruding ledge of shelf". However, these are not discriminative for the class and are accordingly assigned low weights by the classifier. For instance, this could be an image of a wet bar in a resort, however wet bars are also present in places that are not a resort.

3. In Figure 10c, our model correctly predicts "Baseball Field". The prediction is explained by the present of positive concepts like "an outfielder", "white bases at the corners" and "a cleat" (which is a type of baseball shoe). IP-OMP also identifies "action" as a positive concept; however, that is very generic and is not specific to a baseball field. Accordingly, the linear classifier assigns the smallest positive weight to it. Like previous examples, many of the negative concepts in the sparse code are strongly negatively correlated with a baseball

field. These are "a producer", "a light, airy feel", "a large, flat expanse of snow", "cars being worked on", and "often found near a beach".

4. In Figure 10d, our model incorrectly predicts "Operating Room". The true label is "Clean Room" which is what an operating room is also called [82]. Thus, we believe this is a perfectly reasonable classification given the positive concepts extracted by the sparse code are present in the image and are salient features of an operating room. These are namely "medical equipment", "a person working", "sterile surfaces", and "a prep area".

5. In Figure 10e, our model correctly predicts "Arena, Hockey". The positive concepts found by IP-OMP are "pucks", "pads", "a scorpion" (the Hanover Scorpions and the New Mexico Scorpions are hockey teams), "a Zamboni" (a machine used in ice skating rinks to clean the surface) and "state" (this is perhaps one concept whose reason for positive attribution in the sparse code and the classifier weights isn't clear).

### E.3  CIFAR-10

For this dataset, we set $\tau = 9$, that is, the sparse code $\hat{\beta}_{\text{IP-OMP}}$ for every image has support of size 9. A linear classifier trained on these sparse codes gets about $91.6\%$ accuracy, which is within $2.4$ points of what can be achieved if we run the IP-OMP algorithm long enough. This can be seen from Figure 5, where the test accuracy vs. explanation length curve for CIFAR-10 saturates at about $94\%$ accuracy for relatively large $\tau$ (say $\tau = 30$). Results are provided in Figure 11. Unlike ImageNet and Places365, the sparse code is noisier (incorrectly identified positive concepts) for this dataset and CIFAR-100 since these images are much lower resolution and hence out-of-distribution with respect to the distribution of natural images on which CLIP is trained. We do not describe the explanations for each image case-by-case as in the previous two datasets since they are all of the same nature as the previous two datasets. However, we mention some key points. In Figure 11b, IP-OMP assigns a relatively large coefficient to the concept "a large size". However, frogs are typically considered small animals and this is reflected in the classifier weights which assigns the second lowest positive weight (after "reddish-brown coat") to it. In Figure 11c, "antlers", which is a defining feature of a deer, is the most important positively attributed concept, both in the sparse code and in the classifier weights. Similarly, in the same figure, IP-OMP assigns a positive weight to "a halter" which is a headgear for animals like horses. This is probably due to the fact that at such low resolution, the deer looks similar to a horse. However, the linear classifier assigns a relatively negligible weight to this concept indicating that it corrected for this noisy code provided by IP-OMP. Similarly, in Figure 11d, IP-OMP incorrects assigns a positive weight to "quadruped" which is clearly not what a truck is, but the classifier assigns a negligible weight on that concept, indicating that the "quadruped" concept is not used in its prediction for a "Truck". Finally, in Figure 11e, the concept "a beetle" most likely refers to the car model which looks similar to the car displayed in the picture and not the insect.

### E.4  CIFAR-100

For this dataset, we set $\tau = 9$, that is, the sparse code $\hat{\beta}_{\text{IP-OMP}}$ for every image has support of size 11. A linear classifier trained on these sparse codes gets about $64\%$ accuracy, which is within 7 points of what can be achieved if we run the IP-OMP algorithm long enough. This can be seen from Figure 5, where the test accuracy vs. explanation length curve for CIFAR-100 saturates at about $71\%$ accuracy for relatively large $\tau$ (say $\tau = 30$). Results in Figure 12. As explained in §E.3, since the images in CIFAR-100 are of a low resolution (blurry), the IP-OMP sparse codes sometimes make errors by assigning a positive coefficient to a concept that is not related to the object in the image. For example, in Figure 12a "two large, compound eyes" are assigned a positive coefficient in the sparse code, which is unrelated to the class "Forest". Notice the corresponding classifier assigns a negative weight to this concept. In Figure 12c, IP-OMP assigns a positive weight to "baby possums" which is a negative concept for the label of the image, "Rose". Similarly, in Figure 12e, IP-OMP assigns a positive weight to concepts "greenish-brown color" and "a small, narrow neck"; both are incorrect concepts for label "Keyboard". The corresponding classifier assigns a negative and negligibly (the smallest among all concepts in the sparse code) positive weight to them respectively, indicating they are not very important for the classifier's prediction of a Keyboard with 0.87 probability. Figure 12d illustrates an example in which the prediction is completely wrong. The correct label is "Cockroach". The classifier predicts a trout (which is a type of fish) since the sparse code has a relativeily large positive weight on "aquatic animal", probably because the blue background looks similar to an ocean

at such low resolutions (the CIFAR-100 images are originally of $32 \times 32$ resolution, and then upscaled to $224 \times 224$ resolution). We do not describe the explanations for each image case-by-case as in the previous two datasets since they are all of the same nature as discussed for Imagenet and Places365.

### E.5   CUB-200

For this dataset, we set $\tau = 10$, that is, the sparse code $\hat{\beta}_{\text{IP-OMP}}$ for every image has support of size $10$. A linear classifier trained on these sparse codes gets about $43\%$ accuracy, which is within 32 points of what can be achieved if we run the IP-OMP algorithm long enough. More precisely, a relatively large $\tau$ (say $\tau = 155$) is required on this dataset to achieve an accuracy of about $75\%$, which is the accuracy that is obtained if all the atoms in our dictionary were selected, and a linear classifier was trained on that sparse code (these sparse codes are not really sparse since their support size is 208, the size of the dictionary). Results in Figure 13. We do not describe the explanations for each image case-by-case as in the previous two datasets since they are all of the same nature as the previous two datasets. In Figure 13b, the correct prediction is a "Yellow Breasted Chat" which visually looks very similar to a chestnut-sided warbler, which is shown in Figure 8). Both birds have a greenish-yellow back, a small, sparrow-like body, a white body with gray wings, and black cap and white eyeline. As a result, we conjecture that more concepts are needed to discriminate between the two.

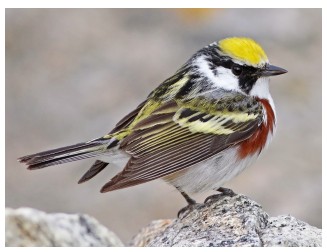

Figure 8: A chestnut-sided warbler, which is the incorrect class that CLIP-IP-OMP predicts for the yellow-breasted chat in Figure 13b.

### E.6   Global Explanations for CLIP-IP-OMP

The examples provided so far were for explaining individual predictions made in terms of the coefficients of the sparse code and the corresponding weights of a linear classifier training on top of these sparse codes. In this subsection we show that one can get a global understanding of the key concepts used in predicting different classes by averaging over the sparse codes generated for that class. Specifically, we consider all test images belonging to a particular class, compute the $\tau$-length sparse codes for all these images. Consequently, we average over these sparse codes to get the mean coefficient for every atom (averaged over all test images belonging to that class). We show these results for three randomly chosen classes from the ImageNet dataset in Figure 14. We see that the top concepts (especially positive ones) for each class are salient features of the respective classes. For example, "Volcano"s are classified by "cone-shaped mountain", "geological phenomenon", "eruption", "crater", "aflame" and so on. "Peru" and "the Andes" are locations where several volcanic mountains are located. Similarly, the top positive concepts (in terms of their mean coefficient value) for "Tiger" is "striped fur", "prey animal", "savannah" (a habitat where tigers are found), "orange and white stripes" and "a fierce expression".

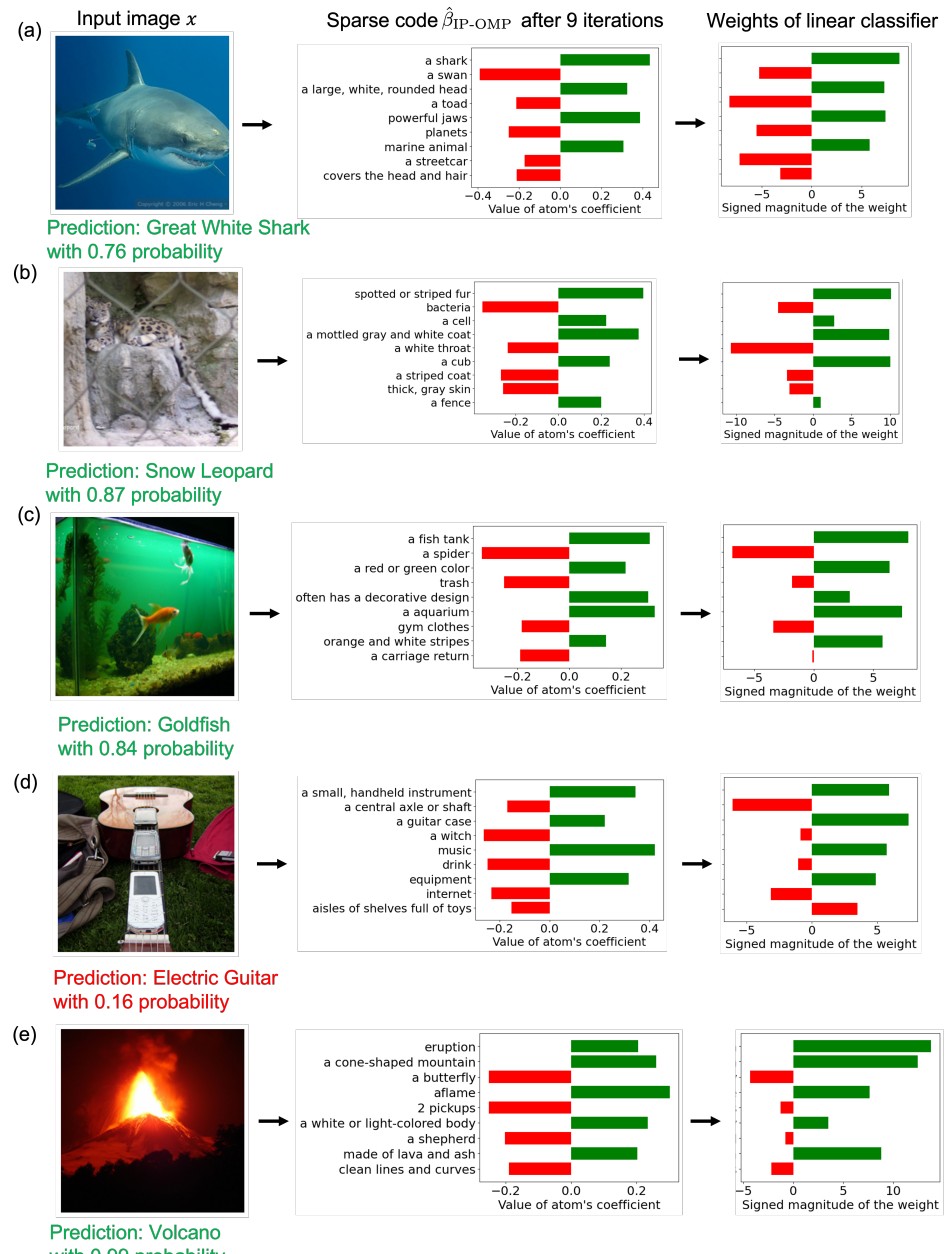

Figure 9: Example runs from the ImageNet validation set. In each row, we display the test image in the first column, the dictionary atoms with non-zero coeficients in the sparse code along with the value of the coefficients in the second column, and finally the weights of the trained linear classifier in the third column. Below every image we note the predicted class. The prediction probability reported is obtained by taking a softmax over the linear classifiers output. The colour scheme is as follows. In column 1, if the prediction is correct it is reported in green, if it is wrong it is reported in red. In column 2, green bars are for positive coefficients and red bars are for negative coefficients. In column 3, we use the same color for the bar as in column 2. For example, if a coefficient was positive, its corresponding weight in column 3 is shown in green regardless of whether the weight was positive or negative.

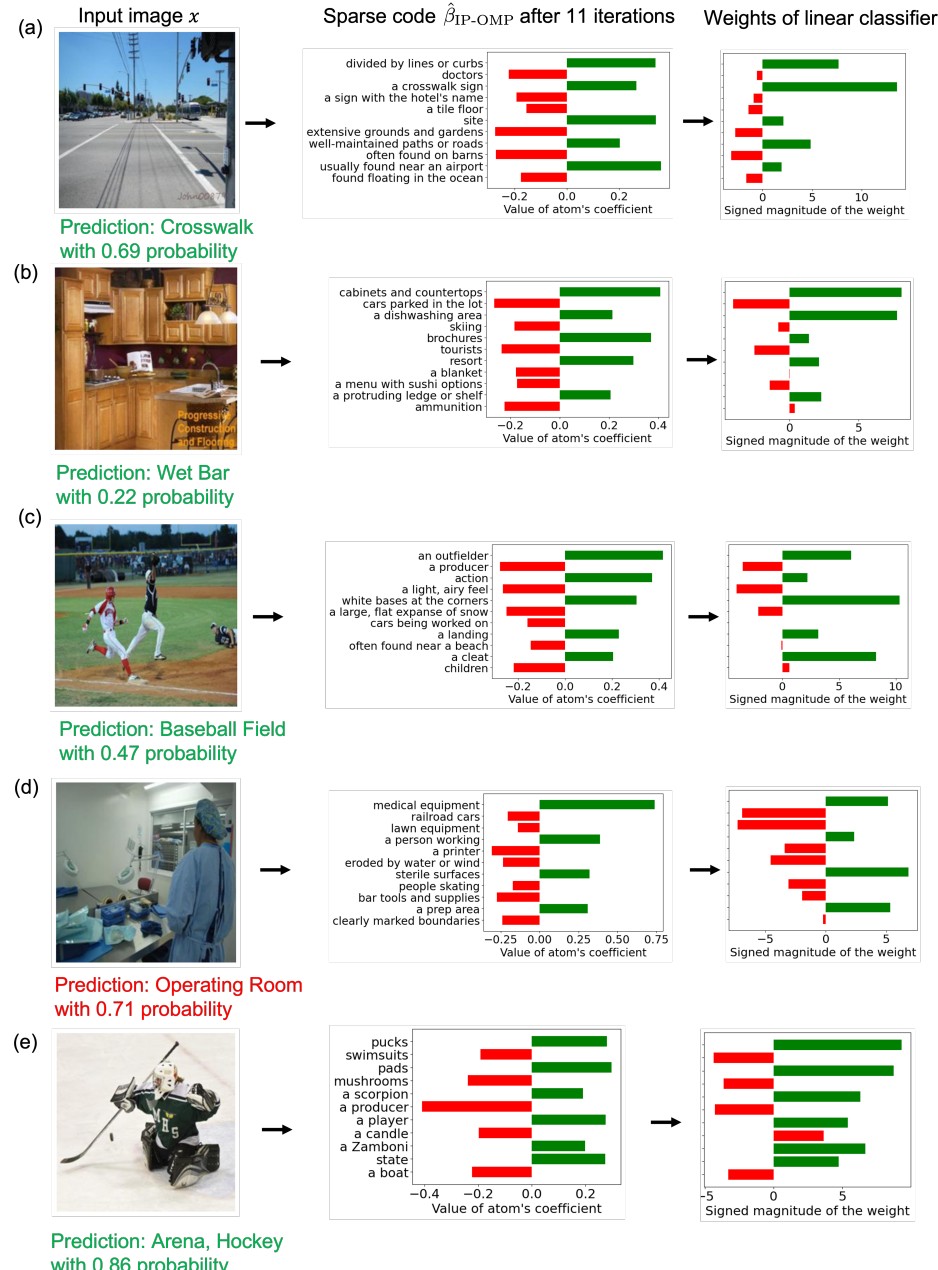

Figure 10: Example runs from the Places365 small validation set. Refer to Figure 9 for a description of the different columns in each row.

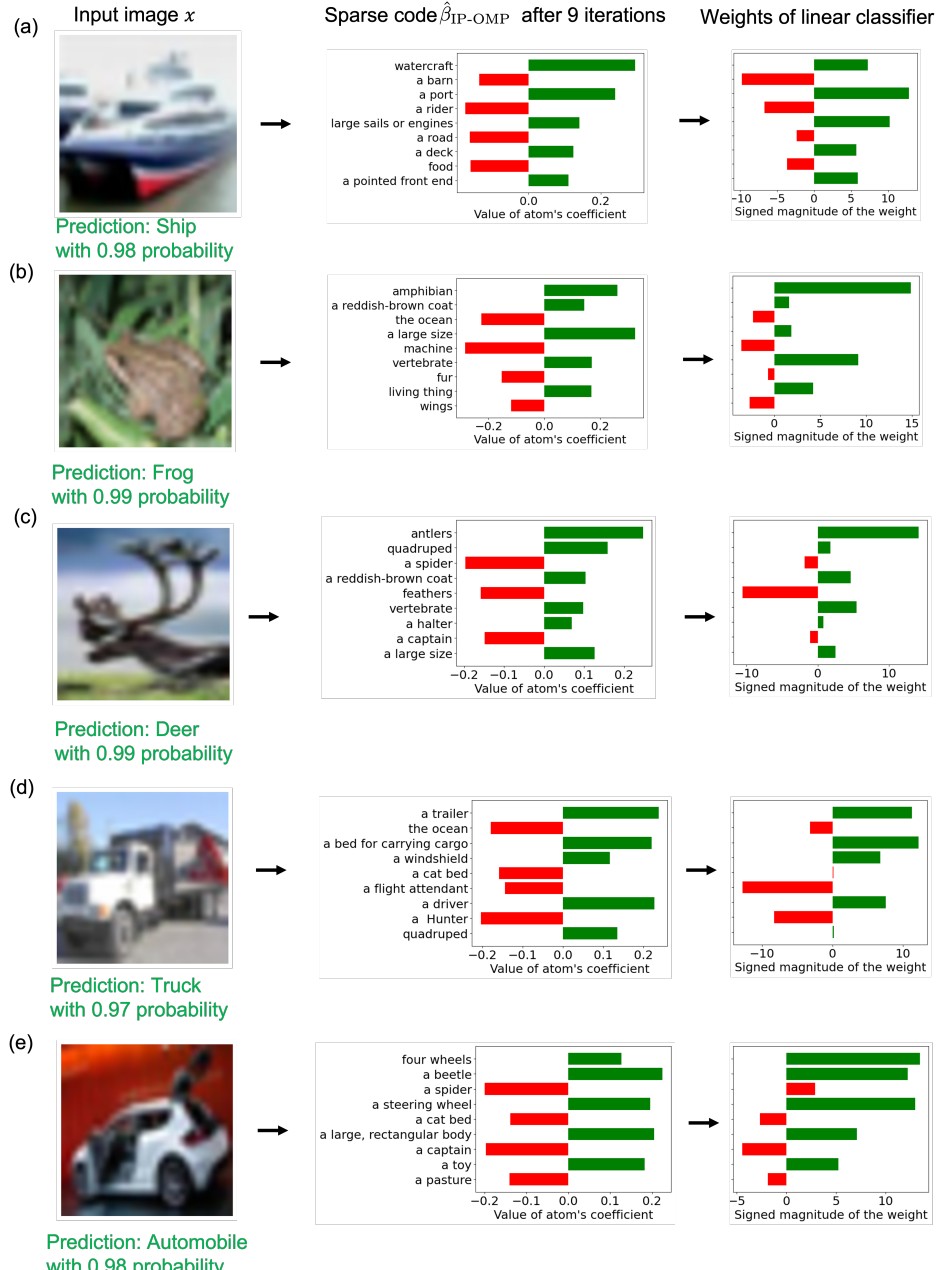

Figure 11: Example runs from the CIFAR-10 test set. Refer to Figure 9 for a description of the different columns in each row.

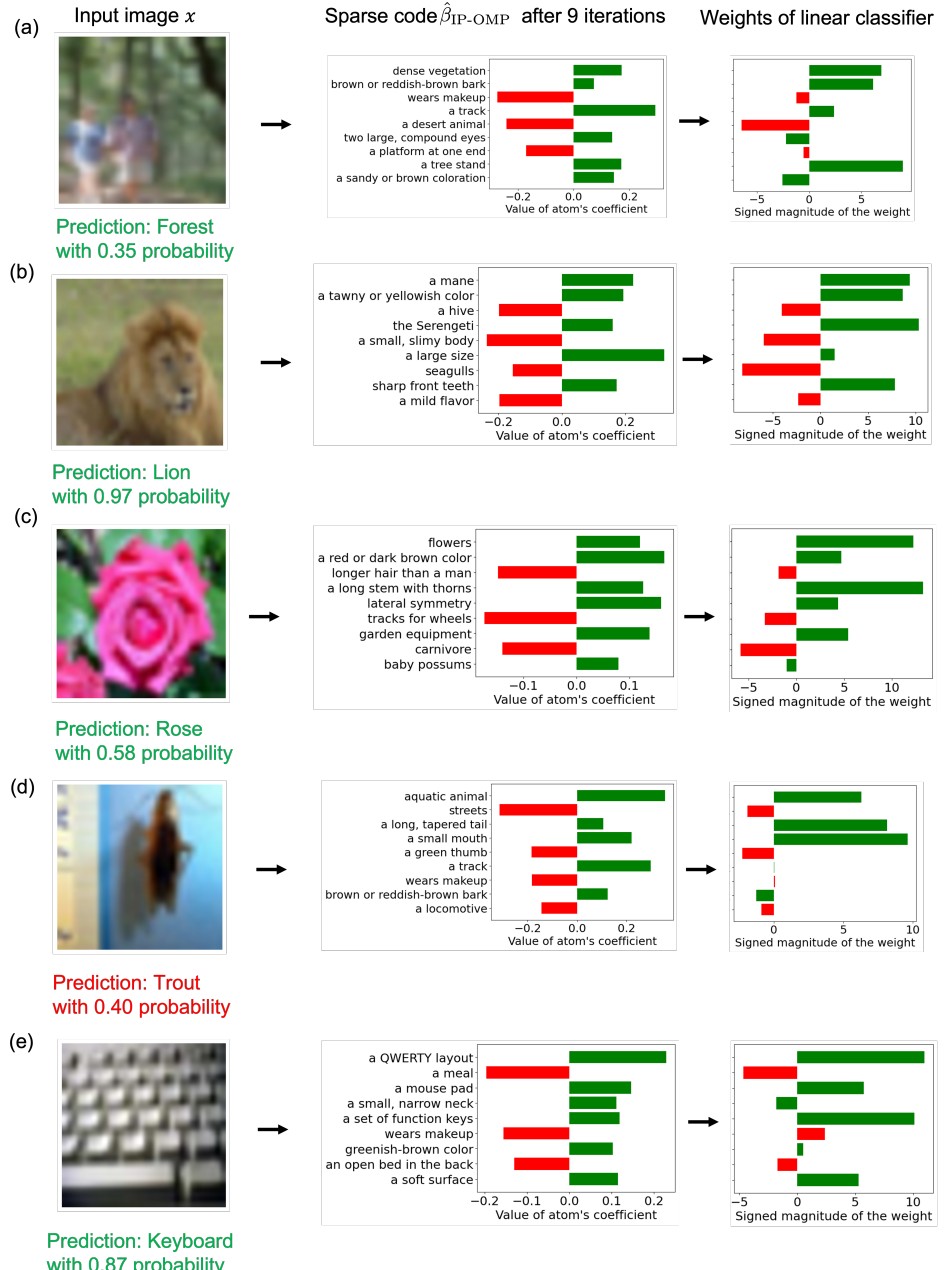

Figure 12: Example runs from the CIFAR-100 test set. Refer to Figure 9 for a description of the different columns in each row.

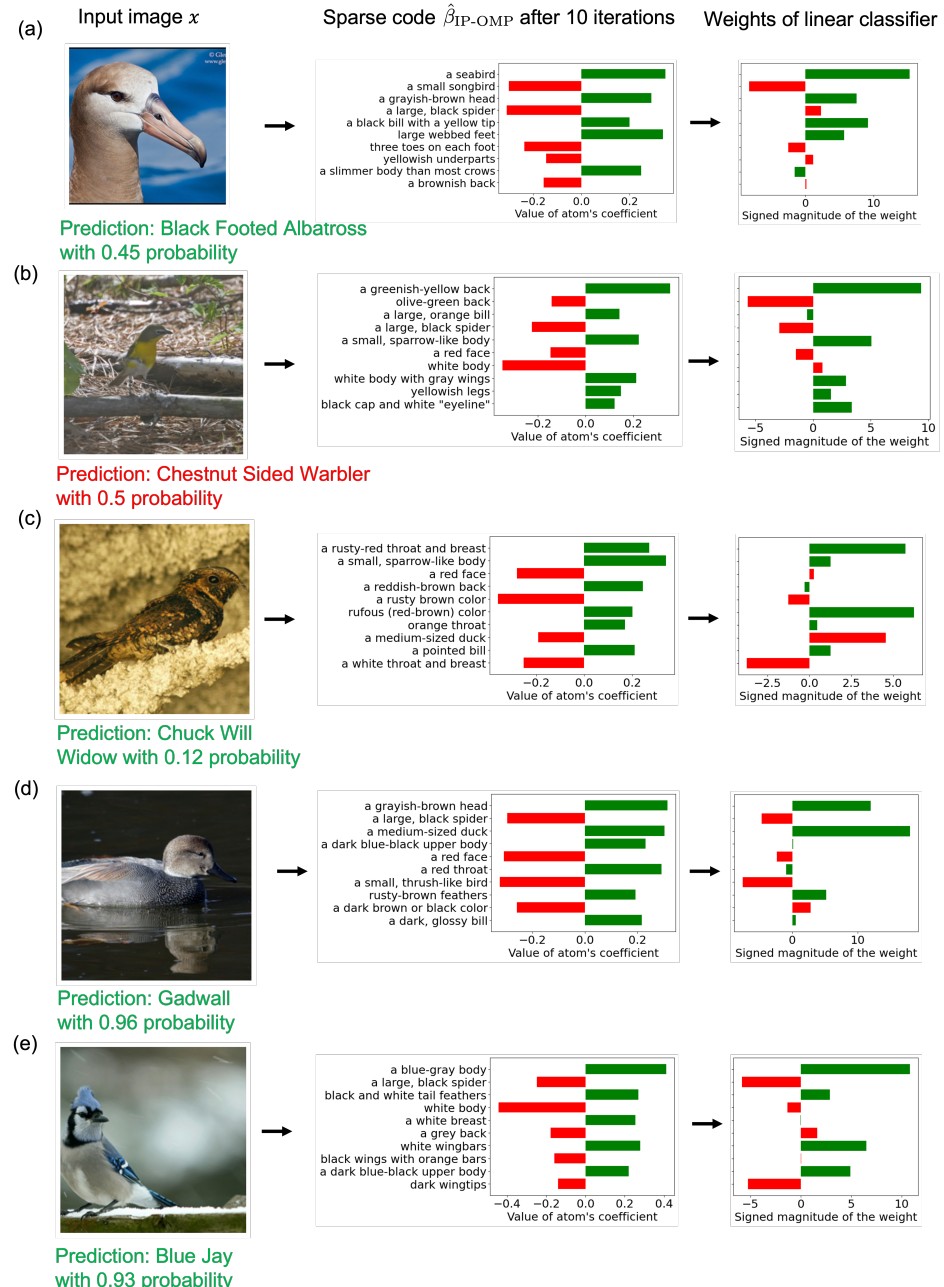

Figure 13: Example runs from the CUB-200 test set. Refer to Figure 9 for a description of the different columns in each row.

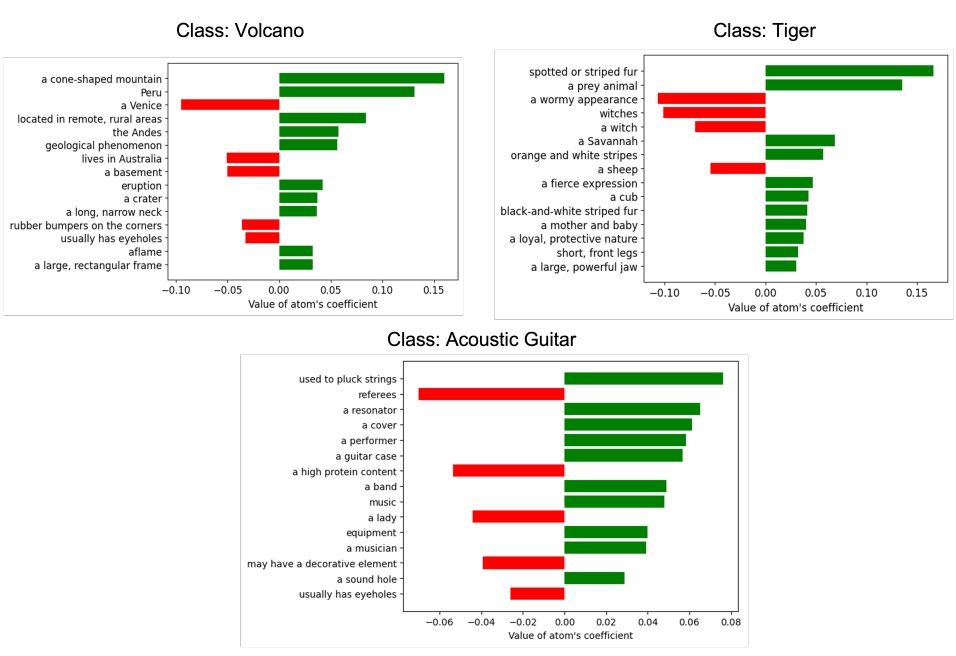

Figure 14: Top 15 concepts identified by CLIP-IP-OMP for different classes from the Imagenet dataset. Each bar shows the average value of coefficients for that concept, averaged over all test images from the same class.

