# OpenReview forum: "Information Maximization Perspective of Orthogonal Matching Pursuit with Applications to Explainable AI"
_NeurIPS.cc/2023/Conference — NeurIPS 2023 spotlight_

### Official Review · Reviewer_tdY5 · 2023-07-02

**Soundness:** 4 excellent
**Presentation:** 3 good
**Contribution:** 4 excellent
**Rating:** 8
**Confidence:** 5

**Summary:**

This paper establishes an intriguing connection between the novel method of Information Pursuit and the well-known approach of Orthogonal Matching Pursuit, foremost used in Compressed Sensing. It then uses this connection for a novel approach to explainability, namely to introduce an approach for visual classification based on queries which is then automatically interpretable. Numerical experiments show the applicabiliy of the approach.

**Strengths:**

* Explainability is an extremely relevant topic.
* The connection between IP and OMP is itself very interesting and revealing.
* The novel explainability approach is elegant and supported by theoretical foundations.
* The numerical experiments are well-thought-through and professionally set up.

**Weaknesses:**

* I am missing an extensive noise treatment, such as when comparing IP with OMP. Right now the original signal seems to be considered a clean signal.

**Questions:**

I don't have any specific questions.

**Limitations:**

I would like to refer to the weaknesses, which are all points, which to my mind need to be addressed and improved.

---

> ### Author Rebuttal · Authors · 2023-08-09
>
> We thank the reviewer for their feedback. Following are our responses to each individual comment (which are highlighted in italics).
>
> > *I am missing an extensive noise treatment, such as when comparing IP with OMP. Right now the original signal seems to be considered a clean signal.*
>
> Thank you for this suggestion. In response to this criticism (Reviewer VN2v also had the same critique) we have carried out synthetic experiments for the noisy case when the true signal is corrupted by Gaussian noise. Please see the PDF attached to the global response for the results for different Signal-to-Noise Ratio (SNR) values. Moreover, the addition of noise does not change the joint Gaussianity of the query and target random variables defined in L182-185 of the main paper. This joint Gaussianity is used in the proof of Theorem 1, which states the equivalence between OMP and IP-OMP (up to a normalization factor). Therefore this result still holds for the corrupted signal.
>
> **Experiment details:** Recall that in the paper we simulated random dictionaries $D \in \mathbb{R}^{m \times n}$ and $s$-sparse codes $\beta \in \mathbb{R}^n$, which generated signals $x$ according to $x = D\beta.$ We extended this setup to the measurement model $x = D\beta + e$, where $e$ is independent Gaussian noise, $D$ is chosen with independent uniform spherical columns, and $\beta$ is generated according to the Gaussian nonzero entry setup described in Appendix B. For each of the $(m, n, s)$-settings of our original noiseless experiments, we simulated noisy measurements with expected SNR values of 5, 10, 15, and 20 dB, where the expected SNR is defined as
> $$
> \mathbb{E}[\mathrm{SNR}] = \mathbb{E}_{D,\beta,e} \left[\frac{\\|D\beta\\|_2^2}{\\|e\\|_2^2}\right].
> $$
> For each trial (draw of $D$, $\beta$, and $e$), we ran OMP and IP-OMP for $s$ iterations each, as in the noiseless experiment. The estimation performance of the two algorithms is reported through plots in the accompanying PDF which effectively show that the equivalence between OMP and IP-OMP is maintained under different levels of measurement noise.

---

> > ### Comment · Reviewer_tdY5 · 2023-08-12
> > **Thank you**
> >
> > Thank you very much for your carefullly written rebuttal and the additional noise analysis.

---

### Official Review · Reviewer_VN2v · 2023-07-07

**Soundness:** 3 good
**Presentation:** 3 good
**Contribution:** 2 fair
**Rating:** 7
**Confidence:** 4

**Summary:**

The paper explores the connection between the information pursuit (IP) approach for active testing and the orthogonal matching pursuit (OMP) algorithm for sparse approximation. By including a normalization step in OMP and posing a statistical model for the queries made in OMP during each algorithm iteration (e.g., projections), OMP can be motivated as an instance of IP. The paper also considers the application of OMP in explainable AI, where semantic attributes and images are embedded into a joint space via the popular CLIP embedding, and the image decomposition via OMP with an attribute dictionary is used as a feature for explainable supervised learning.

**Strengths:**

The presentation for both methods is clear and the framework that draws connections between them is also clearly stated.

The statistical framework for OMP provides a reading of its steps that matches IP once normalization is added. This is not a significant change, as many practical implementations of OMP perform column normalization for the sake of simplicity. Note that the subspace identified by OMP is not affected by the norms of the columns that span it.

The clever application of OMP with CLIP embeddings of semantic concepts opens some interesting questions, in particular whether CLIP embeddings feature a structure similar to that of sparse signals (e.g., concentrated as a union of low-dimensional subspaces).


**Weaknesses:**

The setting of the two algorithms is fairly different, and the mismatch between these settings makes the feasibility of the relationship suffer a bit. In IP, target and query random variables are defined, and mutual information and conditional MI are used to select new queries in a series. In contrast, OMP relies on concepts from vector spaces and high dimensional geometry to estimate components of a decomposition of a signal vector, essentially assuming that the signal of interest is contained within a K-dimensional subspace of the signal's ambient space.

While the normalization added does not change the solution of the problem for exactly sparse signals, there are other considerations regarding noise and approximate sparsity, but those do not appear to be considered in the manuscript. In other words, the study of OMP is limited to exactly sparse signals without noise added.

It is not clear what new insights are brought out by the proposed connection. OMP is a well studied algorithm that has been in the literature for close to 30 years, and column normalization is a standard additional step when applied in practice. Is there any new insight to IP that can be obtained from this connection?

**Questions:**

I wonder whether there is a way to establish a stronger connection by linking two vector spaces, one for each setting. For example, could the IP framework be specialized to RVs that live in a vector space, such as Gaussians?

Line 52: what are "the true optimal solutions" in each case? It is clear for OMP since one is assuming that a single sparse approximation of the signal exists, but it is less clear in the IP case.

Equation 2 is missing "beta hat = arg".

While OMP is an approximation to (2), is IP an approximation? Or is it approximated?

In the second line of IP-OMP, can the second argmax be written as going over the set $D\D_k$?

In line 252, how small does the normalized error need to be for exact recovery? E.g., $10^{-2}$, $10^{-3}$?

In Section 4.1, can the authors provide more information about the linear classifier? How many classes involved? What images are used for training and testing? Are the inputs always the sparse approximation obtained from OMP for the particular image? How is the OMP dictionary built?

In line 319, the reference to Figure 4.2 is incorrect.

In line 414, why is the variance independent of the values that $Q_j$ takes?

In lines 455-456, should P be $P_k$?

**Limitations:**

Limitations are considered in Section 5.

---

> ### Author Rebuttal · Authors · 2023-08-09
>
> We thank the reviewer for their feedback. Following are our responses to each individual comment (which are highlighted in italics).
>
> > *The setting of the two algorithms is fairly different, and the mismatch between these settings makes the feasibility of the relationship suffer a bit. In IP, target and query random variables are defined, and mutual information and conditional MI are used to select new queries in a series. In contrast, OMP relies on concepts from vector spaces and high dimensional geometry to estimate components of a decomposition of a signal vector, essentially assuming that the signal of interest is contained within a K-dimensional subspace of the signal's ambient space.*
>
> We agree with the reviewer that the two algorithms seem to be fairly different in their settings. However, it is not clear why the reviewer feels this makes the feasibility of the relationship between the two algorithms suffer since we theoretically show in this paper that with a specific choice of query and target random variables, it is possible to recover the OMP algorithm (up to a normalization factor) via the IP algorithm. Furthermore, we show both theoretically and empirically (via synthetic experiments) that this normalization factor does not matter for exact recovery of the sparse signal (in the case of no measurement noise).
>
> > *While the normalization added does not change the solution of the problem for exactly sparse signals, there are other considerations regarding noise and approximate sparsity, but those do not appear to be considered in the manuscript. In other words, the study of OMP is limited to exactly sparse signals without noise added.*
>
> Please refer to the response to Reviewer tdY5 for our reply to this comment, since both reviewers had the same criticism.
>
> > *It is not clear what new insights are brought out by the proposed connection. OMP is a well studied algorithm that has been in the literature for close to 30 years, and column normalization is a standard additional step when applied in practice. Is there any new insight to IP that can be obtained from this connection?*
>
> First, we would like to clarify that at any iteration $k+1$, the column normalized implementation of OMP divides the objective in Eq 4 by $\\|d^j\\|$ (for atom $d^j$), whereas IP-OMP divides the same objective by $\\|\Pi_{D_k}^\perp d^j\\|$ (as evidenced by Eq. IP-OMP). This could result in different atom selections by the two algorithms in any particular iteration.
>
> Second, the connection established between IP and OMP opens the door to applying ideas from compressed sensing and sparse signal processing to applications where IP is widely used. This is computationally alluring since OMP is a much simpler algorithm involving vector dot products than IP which requires estimating mutual information in high dimensions (a daunting task). One concrete application in our paper is an explainable AI algorithm for image classification called, CLIP-IP-OMP (see L96-113 in the main paper for more details).
>
> > *I wonder whether there is a way to establish a stronger connection by linking two vector spaces, one for each setting. For example, could the IP framework be specialized to RVs that live in a vector space, such as Gaussians?*
>
> Great question! Yes, the IP framework can be specialized where the target is a Gaussian vector and queries are also Gaussian random vectors correlated with the target. However, it is not clear that this would not result in a stronger connection with OMP since the first query in IP always depends on the joint distribution of the target and query answers (and not the observed signal $x$) whereas the first atom selected by OMP depends on $x$. A similar discussion can be found in L160-176 in the paper.
>
> > *L52: what is "the true optimal solutions" for IP? While OMP is an approximation to (2), is IP an approximation? Or is it approximated?*
>
> Thank you for this question. IP is a greedy approximation to the optimal strategy (the true optimal solution) that minimizes the average number of queries needed to predict $Y$ with a given level of confidence. This has been mentioned in L128-131 of the main paper. However, the reviewer's comment suggests that this clarification should be made prior in the Introduction (before L52) and we will make this change.
>
> > *Eq 2 is missing "beta hat = arg".*
>
> We disagree since Eq 2 states the optimization problem that is solved for recovering the sparse code whereas "beta hat = arg" seems to be referring to the minimizer of the same optimization problem.
>
> > *In the second line of IP-OMP, can the second argmax be written as going over the set $D\D_k$?*
>
> Yes, they are equivalent and it's a matter of taste.
>
> > *In L252, how small does the normalized error need to be for exact recovery?*
>
> We used a threshold of $10^{-14}$ (See Appendix L543).
>
> > *In Section 4.1, can the authors provide more information about the linear classifier?*
>
> We perform our experiments on 5 standard image classification datasets (L312-314) namely: Imagenet, Places365, Cifar-{10,100} and CUB-200. All these datasets come with standard train and test splits which were used for training and testing, respectively. The number of classes for the linear classifier depends on the dataset, for e.g., Imagenet has 1000 classes. Inputs to the classifier are always IP-OMP's sparse code for a particular image (L294-296). Details of the OMP dict. can be found in Appendix L613-646 and further training details can be found in Appendix L648-664.
>
> > *In L319, the reference to Fig. 4.2 is incorrect.*
>
> We apologize for the typo; this should be Fig. 3.
>
> > *In L414, why is the variance independent of the values that $Q_j$ takes?*
>
> This is a property of conditioning in jointly Gaussian variables. If $X$ and $Y$ are jointly Gaussian, the conditional variance of $X$ given $Y=y$ depends only on $Cov(X,Y), Var(X)$ & $Var(Y)$.
>
> > *In L455-456, should P be $P_k$?*
>
> Yes, we apologize for this typo.

---

> > ### Comment · Reviewer_VN2v · 2023-08-11
> > **Thank you for the clarifications.**
> >
> > Some reactions:
> >
> > I concede that "the strength of the connection" is a matter of opinion. In my view, relating IP and OMP because the algorithm can be made to match each other could be strengthened if the settings where the algorithms are applied can also be related to one another, and I think that's not currently present in the narrative given the very different settings where the algorithms are applied. In addition, IP has to be restricted to a specific setting for the match to exist.
> >
> > Regarding normalization, it is common to normalize the inner product in OMP by the norm of the "dictionary vector" involved (e.g., the one that is not signal-dependent). The difference mentioned by the authors has to do with whether the dictionary is manipulated via orthogonal projection or not, a step that is not mandatory in OMP given that it is being applied to the signal anyway. In other words, the projection does not need to be applied to both inputs of the inner product.
> >
> > Regarding equation 2, the sentence before the equation mentions finding the sparse vector beta. Thus, my suggestion would be to phrase (2) in a way that gives such a sparse vector back. This is not a big deal since eq. 3 does give such a description, but I think my suggestion would better set (3) up.
> >
> > Other points in the rebuttal are well made and I have revised my rating accordingly.

---

> > > ### Author Response · Authors · 2023-08-15
> > > **Thank you**
> > >
> > > Thank you for taking the time to respond to our rebuttal. We are pleased with this positive re-evaluation of our work.

---

### Official Review · Reviewer_osjG · 2023-07-13

**Soundness:** 3 good
**Presentation:** 4 excellent
**Contribution:** 3 good
**Rating:** 6
**Confidence:** 4

**Summary:**

This paper analyzes IP in active testing and OMP in sparse signal processing, and establishes connections between the two algorithms. Specifically, IP with a specific configuration can be reduced to OMP up to a normalization, called IP-OMP, and has similar performances as OMP. The paper also proposes to use the IP-OMP algorithm to reduce the computational complexity of IP, and demonstrates its efficiency improvement over variational IP in explainable visual classification,

**Strengths:**

- Provides a novel perspective for bridging two existing techniques in different domains.
- Analysis is sound and creative.
- The presentation of the paper is very clear.

**Weaknesses:**

- As the authors acknowledged, the proposed IP-OMP is limited by the need to place signals and queries into the same domain, which is an intrinsic limitation of the proposed method.
- It would be interesting to see the performance of the proposed IP-OMP on same-modality query problems, such as words-only or images-only, which can eliminate the CLIP module and better demonstrate the performance of IP-OMP itself.

**Questions:**

Please see Weaknesses above

**Limitations:**

The authors addressed the limitations.

---

> ### Author Rebuttal · Authors · 2023-08-08
>
> We thank the reviewer for their feedback. Following are our responses to each individual comment (which are highlighted in italics).
>
> > *As the authors acknowledged, the proposed IP-OMP is limited by the need to place signals and queries into the same domain, which is an intrinsic limitation of the proposed method.*
>
> We agree with the reviewer that this is a limitation of IP-OMP, but emphasize that multimodal models like CLIP allow bridging signal and query domains.
>
> > *It would be interesting to see the performance of the proposed IP-OMP on same-modality query problems, such as words-only or images-only, which can eliminate the CLIP module and better demonstrate the performance of IP-OMP itself.*
>
> Thank you for this suggestion. We would definitely explore this in future work.

---

> > ### Comment · Reviewer_osjG · 2023-08-20
> > **Thank you**
> >
> > I appreciate the authors' response. I will keep my score.

---

### Official Review · Reviewer_FGVf · 2023-07-20

**Soundness:** 3 good
**Presentation:** 3 good
**Contribution:** 3 good
**Rating:** 5
**Confidence:** 4

**Summary:**

The submitted paper proposes information pursuit based orthogonal matching pursuit (IP-OMP) as an explainable classifier for visual classification tasks. IP-OMP is equivalent to conventional OMP with the exception of the normalization. Roughly speaking, IP-OMP uses the standard inner (dot) product $\tilde{a}^{T}\tilde{b}$ after the L2-norm normalization $\tilde{a}=a/\|a\|$ and $\tilde{b}=b/\|b\|$ while OMP only uses the standard inner product $a^{T}b$. The overall operation is not an inner product since it does not have the linearity, i.e. $\tilde{a}^{T}\tilde{b}\neq \tilde{c}^{T}\tilde{b} + \tilde{d}^{T}\tilde{b}$ for $a=c + d$. In this sense, IP-OMP is not a special case of OMP on another inner product space. IP-OMP is derived as a special case of IP, as proved in Theorem 1. The submitted paper follows [37] to utilize CLIP [17] in generating a dictionary for visual classification tasks, so that the proposed algorithm is called CLIP-IP-OMP. Numerical experiments show that CLIP-IP-OMP outperforms V-IP [15] and Lf-CBM [37] in terms of the accuracy in classification for some cases.

**Strengths:**

--Originality

The submitted paper proposes a novel derivation of IP-OMP (i.e. Theorem 1), which is formulated as a special case of IP. While the proof itself is based on elementary techniques in projection and Gaussian estimation, the proposed formulation of IP is not trivial.

--Quality

The proof is probably correct while I did not check detailed computation in Gaussian estimation or projection. At least the main idea to connect IP with OMP is good.

--Clarity

The manuscript is well written and presents a nice review of the background on IP and OMP.

--Significance

The submitted paper reveals the relationship between IP and OMP for applying an OMP-type greedy algorithm to explainable classification.

**Weaknesses:**

My main concern is that CLIP-IP-OMP is questionable as explainable AI: CLIP-IP-OMP is a greedy algorithm selecting inappropriate queries from a human point of view. For instance, Fig. 3 shows that CLIP-IP-OMP selects "a salty flavor", "a witch," "baby birds," and "a strap on the side" to classify a tiger. In my opinion, nobody selects such inappropriate queries to identify a tiger. Numerical experiments seem to show that the selection of such inappropriate queries is a general phenomenon in CLIP-IP-OMP.

The authors claim that such inappropriate queries have negative coefficients (elements of $\hat{\beta}$). The observation itself is supported by numerical evidence. However, if they intend to claim that the negative coefficients indicate inappropriate queries, I disagree with such a leap in logic. From the formulation of the sparse code, inappropriate queries should have coefficients with small magnitude (ideally zero). The sign of the coefficients should not have any meaning unless CLIP encodes an inappropriate query into $-a$ when an appropriate query is mapped into a vector (atom of the dictionary) $a$. It is of course impossible for CLIP to perform such encoding because CLIP does not know the input image "tiger", which determines whether a query is appropriate or not.

In my understanding, the selection of inappropriate queries is an inevitable phenomenon in greedy algorithms, which have no options to discard inappropriate queries selected in early iterations. It is not fair to compare exlainable AIs only in terms of the accuracy in classification, as considered in the submitted paper. It might be possible to improve the accuracy by adding uninterpretable queries (for instance, conventional AI achieves a higher accuracy than humans by utilizing uninterpretable wieghts in a neural network), so that it is possible to compare exlainable AIs in terms of the accuracy only when they do not use inappropriate queries. V-IP [15] and Lf-CBM [37] seem to be in this case.


**Questions:**

--A typo in Theorem 1

$d_i$ in Theorem 1 should have the superscript, i.e. $d^i$.

--page 8, line 319

Is not Figure 4.2 a typo of Figure 3?

--Filtering inappropriate queries

Why do not the authors consider an option to remove inappropriate queries as pre-processing for using CLIP-IP-OMP?

**Limitations:**

No problems.

---

> ### Author Rebuttal · Authors · 2023-08-09
>
> We thank the reviewer for their feedback. Following are our responses to each individual comment (which are highlighted in italics).
>
> > *My main concern is that CLIP-IP-OMP is questionable as explainable AI: CLIP-IP-OMP is a greedy algorithm selecting inappropriate queries from a human point of view. For instance, Fig. 3 shows that CLIP-IP-OMP selects "a salty flavor", "a witch," "baby birds," and "a strap on the side" to classify a tiger.*
>
> We disagree that queries that are not present in an image, like "baby birds" or "a witch" which are not present in the image of a tiger in Fig. 3a, are inappropriate. Absence of concepts can provide discriminatory information for classification. The explainability of our classification algorithm is that of a linear model trained on interpretable features. In our algorithm, each feature corresponds to a semantic concept (like "striped fur") and is thus interpretable. The contribution of a feature to the prediction can be explained by the linear classifier's weight for that feature and the feature's value (the coefficients of the sparse code in this context). This is exactly what is shown in Fig. 3a, features like "a black and white striped fur" have an overall positive contribution to the score for class "Tiger" since both the feature value and the corresponding weight is positive. Similarly, concepts which are anti-correlated with the image content like "a witch" also affect the final prediction score positively since both their feature value and the corresponding weight are negative. This has been discussed in L324-334 in the main paper, and we would modify the writing to emphasize more on this point.
>
> > *The authors claim that such inappropriate queries have negative coefficients (elements of $\hat{\beta}$). The observation itself is supported by numerical evidence. However, if they intend to claim that the negative coefficients indicate inappropriate queries, I disagree with such a leap in logic. From the formulation of the sparse code, inappropriate queries should have coefficients with small magnitude (ideally zero). The sign of the coefficients should not have any meaning unless CLIP encodes an inappropriate query into
> $-a$ when an appropriate query is mapped into a vector (atom of the dictionary) $a$. It is of course impossible for CLIP to perform such encoding because CLIP does not know the input image "tiger", which determines whether a query is appropriate or not.*
>
> We disagree with the reviewer. We would like to clarify that we never claim inappropriate queries have negative coefficients. We make the claim that concepts that are assigned negative coefficients in the sparse code by IP-OMP are concepts that are not found in the image. This claim stems from CLIP's training objective which is a contrastive loss that optimizes image and text representations such that if the text aligns with the contents of the image, the normalized dot product between their corresponding representations is maximized (pushed closer to $1$). Similarly, if the given text does not correspond to the image content, the normalized dot product between their corresponding representations is minimized (pushed closer to $-1$). From this perspective, we believe that it is reasonable to expect that if CLIP encodes a text concept (say "a witch") as $a$ and if the image content can be described well by the concept then its representation $x$ would have a high dot product with $a$ and if the concept does not describe the content of an image, it would have a low dot product with $x$ (the lowest being $-1$). This is our rationale for assigning meaning to the coefficients in the sparse code, where the sign indicates correlation (positive or negative) of the image content with the concept and magnitude indicates strength of that correlation.
>
> > *It is not fair to compare exlainable AIs only in terms of the accuracy in classification, as considered in the submitted paper, so that it is possible to compare exlainable AIs in terms of the accuracy only when they do not use inappropriate queries. V-IP [15] and Lf-CBM [37] seem to be in this case. In my understanding, the selection of inappropriate queries is an inevitable phenomenon in greedy algorithms, which have no options to discard inappropriate queries selected in early iterations.*
>
> This is incorrect. We compare explainable AI algorithms via their tradeoff between description length (the number of concepts used in prediction) vs. the accuracy obtained. The description length is a measure of explainability since a long list of concepts would make the predictions less intelligible. If two algorithms have the same average description length we consider them to be equally well explainable since, in our opinion, there is no well-defined metric for ascertaining one explanation to be better than another (when they are based on the same set of interpretable features) beyond its description length. Moreover, we do not agree that the selection of negative queries/concepts (which the reviewer refers to as inappropriate queries) is inappropriate as argued in previous comments.
>
> > *$d_i$ in Theorem 1 should be $d^i$.*
>
> This is not a typo. We use superscripts to indicate column indices within the dictionary matrix, while subscripts indicate the order these atoms are chosen in IP-OMP. So, $d_i$ is the atom selected in iteration $i$, whereas $d^i$ refers to the $i^{th}$ column of the dictionary. This distinction is first made in footnote 1 (and also on L143-144 and L205-206) of the paper.
>
> > *Is Fig. 4.2 a typo of Fig. 3?*
>
> Yes, sorry for the typo; this should be Fig. 3.
>
> > *Why do not the authors consider an option to remove inappropriate queries as pre-processing for using CLIP-IP-OMP?*
>
> As argued in previous responses, we do not agree with the reviewer that the negative concepts selected by our method are inappropriate.

---

> > ### Comment · Reviewer_FGVf · 2023-08-10
> > **Score update**
> >
> > Thank you for your reply. I have understood why the sign of the coefficients can have a meaning and increased my score slightly. (I do not oppose acceptance.) Nonetheless, I am not convinced that authors’ rebuttal resolves my main concern fully. I believe that this conflict is due to ambiguity in the meaning of words, such as “explainable” and “inappropriate.”
> >
> > I interpreted explainable AI as AI that generates a chain of queries as if a human created them. In this sense, a chain of “appropriate” queries for a tiger would be “predatory animal,” “striped fur,” “jungle,” “loneliness,” and so on. If “a witch” is included in this chain of queries, a human would regard it as an inappropriate query. The meaning of this “inappropriate” might be close to “inefficient” in human’s intuition. (It might be “efficient” in terms of machine learning.)
> >
> > On the other hand, the authors seem to regard explainable AI as AI that generates a chain of queries that are possible for a human to interpret. In this sense, a feature vector in a high-dimensional space is an “inappropriate” query. However, “a witch” is an “appropriate” query for a tiger because a human can interpret its meaning.
> >
> > I still believe that the former human-like AI should be a ultimate target in future research and that greedy algorithms tend to generate inappropriate queries in terms of the human-like AI. On the other hand, inappropriate queries are never generated in terms of the latter AI because the set of queries only contains appropriate (interpretable) queries. I hope that the meaning of “explainable” is clarified in revision to circumvent a confusion.

---

> > > ### Author Response · Authors · 2023-08-15
> > > **Thank you**
> > >
> > > Thank you—we appreciate your taking the time to read and respond to our rebuttal. We will certainly clarify what is meant by “explainable AI” in the paper based on this discussion.

---

### Author Rebuttal · Authors · 2023-08-10

We thank all the reviewers for their time and insightful comments which have helped improve our paper. We are elated that the reviewers found our paper well-written, the presentation clear and our connection between IP and OMP as original, creative and revealing. We are pleased that the reviewers found our application of IP-OMP to explainable AI for image classification using CLIP embeddings as interesting, novel and extremely relevant. Following are some of the main critiques, afterwards we address each reviewer's comments individually.

1. Reviewer FGVf had concerns over the explainability of our proposed CLIP-IP-OMP algorithm. We believe we have addressed these concerns in our detailed response to the reviewer and would incorporate a part of this discussion into our paper.

2. Reviewer osjG provided interesting suggestions for applying our algorithm to scenarios where query and signal domains are the same, such as words-only or images-only. We would certainly explore this direction in future investigations into the IP-OMP algorithm.

3. Reviewer VN2v had concerns about the disparate settings of the two algorithms, namely IP deals with random variables and uses mutual information to select the next query, whereas OMP relies on concepts from vector spaces and high dimensional geometry. We believe that this disparity makes our contribution, which shows that OMP can be derived from IP by a specific choice of query and target random variables, all the more surprising and exciting.

4. Reviewer tdY5 and VN2v were missing a thorough treatment of noise in the observed signal (measurements). In response to this, we have conducted synthetic experiments at different noise levels, number of measurements, number of dictionary atoms, and sparsity levels of the signal. In all experiments, we see the performance of IP-OMP and OMP to be almost the same, indicating that the equivalence between the two algorithms is also preserved under noise. These results would be added to the appendix of the paper.

---

### Decision · Program_Chairs · 2023-09-21

**Decision:**

Accept (spotlight)

**Comment:**

The paper draws connections between Information Pursuit, a somewhat complex greedy algorithm recently used in producing explainable classification decisions, and Orthogonal Matching Pursuit, a classical greedy algorithm for sparse approximation. The paper shows that OMP can be seen as a particular case of IP, in which the ``questions’’ greedily selected by IP correspond to atoms in a dictionary. Based on this, the paper proposes OMP as a computationally efficient alternative to IP, and demonstrates promising results on explainable visual classification. Reviewers found the paper to be technically solid and clearly written, with new insights into greedy algorithms, and promising results on visual classification. After some discussion around the notion of explainability offered by the paper (and its limitations), reviewers converged to a recommendation to accept the paper.